# *BrainFlow*: A Holistic Pathway of Dynamic Neural System on Manifold

**Zhixuan Zhou**          **Tingting Dan**          **Guorong Wu**[*]
Departments of Computer Science and Psychiatry
University of North Carolina at Chapel Hill
Chapel Hill, NC 27599
{Tingting_Dan,grwu}@med.unc.edu;zzhixuan@cs.unc.edu

## Abstract

A fundamental challenge in cognitive neuroscience is understanding how cognition emerges from the interplay between structural connectivity (SC) and functional connectivity (FC). Current machine learning approaches typically seek to establish direct mappings from SC to FC associated with specific cognitive states. However, these methods often treat SC and FC as distinct endpoints, failing to capture the coupling relationship throughout the progressive transformation between them. To address this limitation, we propose BrainFlow, a reversible generative model designed to parametrize flows between the distribution of SC and the mixed distribution of FCs from different cognitive tasks. Our method explicitly models the SC-FC coupling by training on interpolated states along the symmetric positive-definite (SPD) manifold. We further prove the equivalence between flow matching on the SPD manifold and on the computationally efficient Cholesky manifold, enhancing numerical stability. To mitigate cumulative errors during reverse-flow simulation, we introduce a consensus control mechanism that utilizes complementary information across multiple FC-to-SC pathways, yielding a biologically meaningful reconstruction of the underlying structural scaffold. Together, BrainFlow achieves state-of-the-art performance on both synthetic data and large-scale neuroimaging datasets from the UK Biobank and Human Connectome Project.

## 1   Introduction

A central goal in cognitive neuroscience is understanding how cognition arises from the complex interplay between (static) brain structure and (dynamic) functional fluctuations. Structural connectivity (SC), representing the anatomical pathways that physically connect distinct brain regions, provides the structural scaffold that supports the spontaneous synchronization of functional fluctuations across multiple brain regions, shaping the network topology of functional connectivity (FC). This interplay underpins the brain's ability to perform complex cognitive, sensory, and motor functions, highlighting the necessity of understanding how anatomical structure constrains and shapes brain function, and vice versa [1, 2, 3].

Machine learning techniques have been widely employed to characterize the uni-directional relationship between SC and FC. One streamlined approach is to simply predict one type of connectome from another in a supervised learning scenario[4, 5, 6].

Despite promising prediction accuracy, these computational approaches have several significant issues, from problem formulation to model explainability. **(1) Over-simplified problem formulation.** Most current methods only focus on the one-way mapping function, i.e., either predicting FC from

---

[*]Corresponding author.

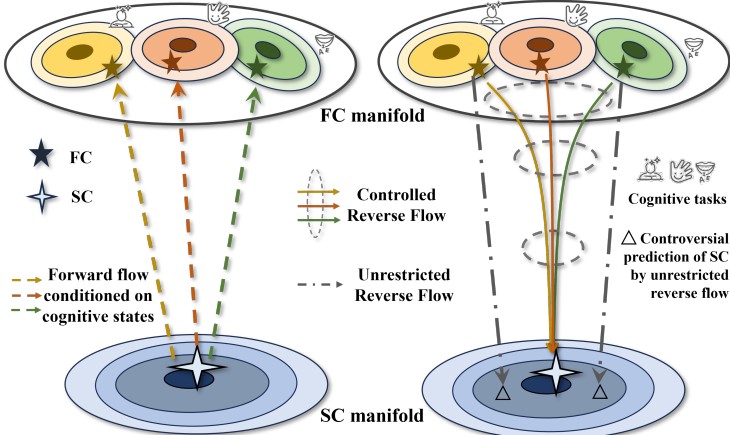

Figure 1: We model the SC-FC relationship in human brain as a flow problem, characterized by a collection of one-to-many forward flows from single (static) SC to multiple (evolving) FCs, along with corresponding many-to-one reverse flows from state-dependent FCs back to the underlying SC. Recognizing the neuroscience insight that a significant portion of neural circuits are shared across different cognitive tasks, we introduce the concept of consensus control to harmonize the reverse flows between cognitive states.

SC or vice versa. However, mounting evidence from neuroscience studies underscores the interplay between SC and FC. In this context, the real biological challenge is establishing a new understanding of SC-FC coupling mechanism in human brain using data-driven approaches. **(2) Overlook the intrinsic data geometry.** It is popular to learn feature representation for SC and FC in a vector space by flattening the region-to-region connectivity matrix into a vector. Since the matrix form of connectome data does not live in a Euclidean space [7], manifold-based algebra tailored for the unique data geometry (such as the symmetric and positive-definite properties) is needed to guarantee the mathematics and neuroscience insight in operating the connectome data.

To address these limitations, we propose *BrainFlow*, a reversible generative model designed to parametrize flows between the distribution of SC and the mixed distribution of FCs on the Riemannian manifold (Fig. 1). Conceptually, a flow in this context is not a direct physical process but a learned transformation pathway, akin to a continuous interpolation, that connects the geometric space of structural connectomes to that of functional connectomes. Specifically, we consider that each SC or FC is a particle sampled from its respective distribution on the manifold. In this context, the relationship between SC and FC can be formulated with a flow governed by a time-dependent ODE.

Following the spirit of trajectory inference [8, 9],we parameterize pathways that transport SC distributions to state-specific FC distributions. This enables generating FC in different states for unseen SC data and, crucially, reconstructing SC from FC without additional training.

We further propose three innovative components to find mapping flows between SC and FCs. **(1)** We present a flow model that not only predicts FCs at different states from SC but also reverses the process to recover SC from FC. This bidirectional capability is essential for understanding the interplay between the brain's structural and functional organization, enabling insights into how anatomical constraints shape functional dynamics and how functional variations reflect structural substrates. **(2)** We parametrize the flow on the *Cholesky* manifold [10], an isometric embedding of the symmetric positive-definite manifold (Fig. 2). This preserves the geometric and probabilistic properties of the original SPD space while avoiding the computational overhead of traditional affine-invariant Riemannian metrics, ensuring efficient and stable operations for high-dimensional matrices. By employing the *log-Cholesky* metric, *BrainFlow* ensures efficient and stable operations while preserving the intrinsic structure of SC and FC data. **(3)** We introduce a consensus control mechanism that leverages the shared structural foundation underlying multiple functional states (demonstrated in Fig. 1 right). By treating SC as a stable anchor and introducing consensus-driven adjustments into the dynamics, *BrainFlow* ensures that predictions across different FC states (e.g., resting-state and task-specific FCs) are both consistent and biologically reasonable.

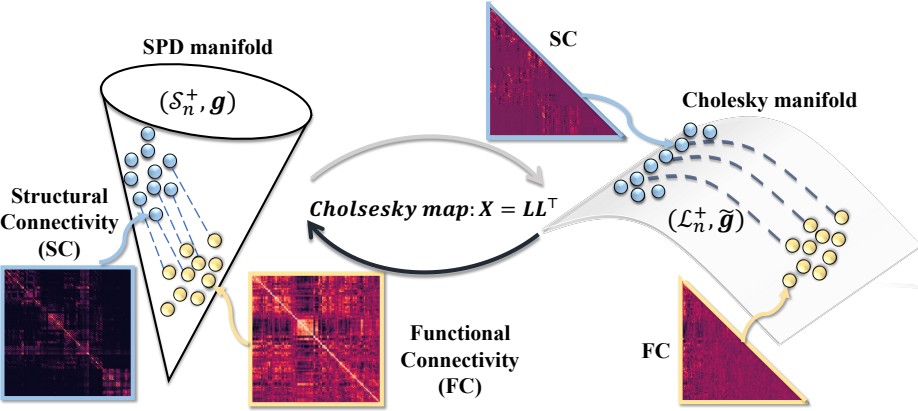

Figure 2: Both SC and FC are regarded as data elements drawn from their respective distribution on the SPD manifold. Rather than estimating the flow directly on the SPD manifold, we perform optimization on the Cholesky manifold, which is computationally efficient while being equivalent to the SPD manifold.

We compare our methods to baselines and large-scale neuroimaging datasets from UK Biobank and Human Connectome Project. In addition to competitive or state-of-the-art performance on different metrics, our method shows faster training speed and more stable inference.

## 2 Preliminaries

### 2.1 Riemannian, SPD, and Cholesky Manifold

**Riemannian manifold.** A Riemannian manifold $(\mathcal{M}, g)$ is a smooth manifold $\mathcal{M}$ equipped with a Riemannian metric $g$ that defines an inner product on each tangent space $\mathcal{T}p\mathcal{M}$. The geodesic $\gamma(t)$ between two points on a Riemannian manifold represents the locally length-minimizing curve, analogous to straight lines in Euclidean space. Two fundamental operations on Riemannian manifolds are the exponential map $\exp(\cdot)$ and its inverse, the logarithmic map $\log(\cdot)$. The exponential map $\exp_p(\cdot) : \mathcal{T}p\mathcal{M} \to \mathcal{M}$ maps a tangent vector $v$ to the point reached by following the geodesic starting at $p$ in direction $v$ for unit time, while $\log_p(\cdot) : \mathcal{M} \to \mathcal{T}p\mathcal{M}$ maps a point $q$ to the tangent vector $v$ such that $\exp_p(v) = q$.

**SPD manifold.** Symmetric positive definite (SPD) matrices form a SPD space that naturally represents functional brain connectivity [7, 11]. When equipped with a Riemannian metrics like commonly used Log-Euclidean metric [12] or affine-invariant metric [13, 14], $\mathcal{S}_n^+$ turns into a Riemannian manifold. However, these classical metrics involve matrix exponential and logarithm operations that are computationally expensive and potentially numerically unstable for large-scale or ill-conditioned matrices due to eigendecomposition.

**Cholesky manifold.** Recently, Lin ([10]) proposed the Log-Cholesky metric, which mainly relies on Cholesky decomposition $X = LL^T$, where $X \in \mathcal{S}_n^+$ and $L$ is a lower triangular matrix whose diagonal elements are positive. Lin ([10]) first introduced the Riemannian manifold of lower triangular matrices, which are the Cholesky decomposition factors of SPD matrices, and proves there exists an *isometry* between the SPD manifold and lower triangular manifold. Here we briefly introduce the *Cholesky space*. We firstly denote $\mathcal{L}_n^+$ as a space whose elements are lower-triangular matrices with positive diagonal. The notation $\lfloor A \rfloor$ denotes the strict lower triangular part of matrix $A$, specifically, $\lfloor A \rfloor_{i,j} = A_{i,j}$ if $i > j$ and zeros otherwise. $\mathbb{D}(A)$ represents a diagonal whose diagonal elements are those of $A$. We then introduce some important properties of Cholesky manifold $(\mathcal{L}_n^+, \tilde{g})$, where $\tilde{g}$ is a Riemannian metric. First, the tangent space $\mathcal{T}_L\mathcal{L}_n^+$ at $\mathcal{L}_n^+$ is $\mathcal{L}_n$, the space of lower triangular matrices. Second, the inner product between $P, Q \in \mathcal{T}_L\mathcal{L}_n^+$, i.e., the Riemannian metric $\tilde{g}$, is defined as

$$\tilde{g}_L(P,Q) = \sum_{i>j} P_{ij}Q_{ij} + \sum_{j=1}^{n} P_{jj}Q_{jj}L_{jj}^{-2}, \tag{1}$$

Lin ([10]) further induced the log-Cholesky metric $g$ on $\mathcal{S}_n^+$. Specifically, given $X \in \mathcal{S}_n^+$ and $W, V \in \mathcal{T}_X\mathcal{S}_n^+$, the *log-Choleksy metric* is defined as:

$$g_X(W,V) = \tilde{g}_L\left(L(L^{-1}WL^{-\top})_{\frac{1}{2}}, L(L^{-1}VL^{-\top})_{\frac{1}{2}}\right), \tag{2}$$

where $L$ is the Cholesky factor such that $X = LL^\top$ and $(\cdot)_{\frac{1}{2}} = \lfloor\cdot\rfloor + \mathbb{D}(\cdot)/2$ for an input square matrix.

Next, the geodesic between two points $K, L \in \mathcal{L}_n^+$ on the Cholesky manifold is defined as:

$$\tilde{\gamma}_{L,K}(t) = \lfloor L \rfloor + t\{\lfloor K \rfloor - \lfloor L \rfloor\} + \mathbb{D}(L)\exp[t\{\log\mathbb{D}(K) - \log\mathbb{D}(L)\}], \tag{3}$$

where the strict lower triangular part can be regarded as the straight line in Euclidean space and the diagonal part is designed to meet the constraint of Cholesky space. More basic properties about Cholesky manifold and SPD manifold can be found in Appendix. A

### 2.2 Flow Matching on Riemannian Manifold

**Riemannian Flow Matching.** Given a flow $\psi_t$ that generates a probability path $\rho_t$ connecting two distributions $\rho_0$ and $\rho_1$, we can know the vector field $u_t$ by $\frac{d}{dt}\psi_t(x) = u_t(\psi_t(x))$. Then we can fit a parametrized vector field $v_t^\theta$ to $u_t$ by optimizing the loss $\mathcal{L}_{RFM} = \mathbb{E}_{t,\rho_t(x_t)}\|v_t^\theta(x_t) - u_t(x_t)\|_g^2, t \sim$ uniform$(0,1)$. This method is named *(Riemannian) Flow Matching* (RFM) [15, 16, 17, 8]. However, $u_t$ is intractable in most cases. Conditional flow matching therefore is then proposed to use $v_t^\theta$ to approximate a conditional vector field $u_t(x_t|z)$ defined on each sample which also generates a conditional probability path $\rho_t(x_t|z)$. Then we can recover the unconditional vector field and probability path by $u_t(x_t) = \int_\mathcal{M} u_t(x_t|z)\frac{\rho_t(x_t|z)q(z)}{\rho_t(x_t)}d\mathrm{vol}_z$ and $\rho_t(x_t) = \int_\mathcal{M} \rho_t(x_t|z)q(z)d\mathrm{vol}_z$ [8]. The Riemannian flow matching objective then becomes

$$\mathcal{L}_{RCFM} = \mathbb{E}_{t,\rho_t(x_t|z),q(z)}\|v_t^\theta(x_t) - u_t(x_t|z)\|_g^2 \tag{4}$$

It has been proved that the loss $\mathcal{L}_{RFM}$ and $\mathcal{L}_{RCFM}$ have same gradients[8] *w.r.t.* $\theta$. We can use the conditional velocity field as target to fit $v_t^\theta$.

### 2.3 Brain Connectivity

The human brain's connectivity can be characterized through two distinct mathematical representations defined on a set of $n$ brain regions $\mathcal{V} = \{v_1, ..., v_n\}$.

Structural connectivity (SC) describes the anatomical architecture through matrix $A^s \in \mathbb{R}^{n \times n}$, where each element $A_{i,j}^s \geq 0$ represents the connection strength between regions $v_i$ and $v_j$. Structural connectivity is typically modeled as an undirected weighted/binary graph.

In contrast, functional connectivity (FC) is represented by $A^f \in \mathbb{R}^{n \times n}$ quantifying temporal relationships, where $A_{i,j}^f$ is the correlation between Blood Oxygen Level Dependent (BOLD) signals of regions $v_i$ and $v_j$. SPD matrices naturally arise in the analysis of functional brain connectivity patterns through its derivation from correlation or covariance measures. Though SC is not guaranteed to be SPD, it is common to satisfy SPD property by adding a (small) self-loop connection to the symmetric SC matrix.

## 3 Related Works

### 3.1 Prediction between Functional Network and Structural Network

There have been many works aimed at predicting functional connectivities from structural connectivities or vice versa. Traditional methods[2, 18, 19] typically used the statistical properties (e.g. connection strength, distances, shortest paths, eigenvalues of Laplacian matrices) of one type of connectivities to predict the other connectivities. However, these methods mainly focused on small-scale or even single subject, ignoring potential common properties among the population. Some

machine learning methods are also used for predicting the connectivity. MGCN-GAN[6] uses a CycleGAN-based network to predict structural connectivities with functional connectivities while DSBGM[5] proposed a signed graph encoder to map functional connectivities to structural connectivities. These methods considered only the one-way mapping and focused only on resting-state functional connectivities.

### 3.2 Transformation between two distributions

Transformation between two distributions is an important problem in natural sciences[20, 21]. Recently, many simulation-free distribution interpolation techniques were proposed[8, 15, 17, 22, 23]. These methods trained a time-dependent ODE to evolve an initial distribution to the target distribution. Most of them worked with the distributions on Euclidean space. RCFM[8] proposed conditional flow matching on Riemannian manifold. Based on RCFM, we further explore flow matching among isometries, which enables applying flow matching on a simpler manifolds and then maps it back to the complex isometry, making the framework more efficient.

## 4 Flow Matching for Bridging Structural and Functional Connectivity

### 4.1 Problem formulation

In this work, we aim to predict resting-state and task-specific functional connectivity (FC) matrices given structural connectivity (SC) matrices from the same individual. In addition, we also aim to predict SC matrices based on given FC matrices. Specifically, we denote the distribution of structural connectivity matrices $A^s$ as the source distribution $\rho_0$, and the distribution of functional connectivity matrices $A^f$ as the target distribution $\rho_1$. Elements sampled from these distributions are represented as $X_0 \sim \rho_0$ and $X_1 \sim \rho_1$, where $X_0$ corresponds to an SC matrix $A^s$ and $X_1$ corresponds to an FC matrix $A^f$.

Our objective is to train a vector field $v_t^\theta$ such that, given an SC matrix $X_0^i$ for subject $i$, the corresponding resting or task-specific FC matrix can be predicted via integration:

$$X_1^i = X_0^i + \int_0^1 v_t^\theta(X_t^i, y^i)\, dt, \tag{5}$$

where $y^i$ is the label of the target FC (e.g., resting-state, motor, or working memory). On the other hand, the method can also predict SC given a labeled FC matrix.

### 4.2 Flow matching on Cholesky Manifold

**Motivation.** Our approach aims to construct a conditional vector field $u_t(X_t|Z) \in \mathcal{T}_{X_t}\mathcal{S}_n^+$ that transports $X_0 \sim \rho_0$ to $X_1 \sim \rho_1$ and use a neural network $v_t^\theta$ to fit $u_t(X_t|Z)$.

**Subject-specific prior conditioner.** Following [17], we define the prior conditioner $Z$ as a joint distribution of source and target: $Z = (X_0, X_1) \sim q(Z) = q(X_0, X_1)$. Due to the individual-specific nature of connectivities, however, we do not employ either independent coupling $q(X_0, X_1) := \rho_0(X_0)\rho_1(X_1)$ or optimal transport coupling $q(X_0, X_1) := \pi(X_0, X_1)$. Instead, we define a specialized joint distribution $q(Z)$ that ensures each sample $Z$ corresponds to an individual. In this formulation, $X_0$ and $X_1$ represent SC and FC matrices derived from the same subject, preserving biologically coherent and individualized relationships between structural and functional domains.

**Construct flow with geodesic.** we then construct a flow $\psi_t$ to connect $X_0$ and $X_1$. Analogous to the straight line in Euclidean, it is natural to use geodesic connecting $X_0$ and

$$X_t = \gamma_{X_0, X_1}(t) = \exp_{X_0}(t\log_{X_0}(X_1)) \tag{6}$$

where the exponential and logarithmic maps are computed under a Riemannian metric on the symmetric positive definite (SPD) space $\mathcal{S}_n^+$.

**Flow matching with *log-Cholesky metrics*.** A natural choice of Riemannian metric on $\mathcal{S}_n^+$ is the affine-invariant Riemannian metric (AIRM), as used in [8]. While AIRM is effective in low-dimensional cases, its computational complexity poses challenges for high-dimensional SPD manifolds due to the reliance on matrix eigendecomposition, which becomes numerically unstable when

eigenvalues approach zero. Additionally, the computations of fractional matrix powers and matrix inverses contribute to high computational costs. (See Appendix B.2 for details.) These factors make AIRM difficult to apply to higher-dimensional SPD manifold. To address these challenges, we adopt the *log-Cholesky metric* in eq. 2, whose main overhead of geodesic and inner product computation is only matrix inverse. Then the objective becomes:

$$\mathcal{L}_{\text{RCFM}}(\theta) = \mathbb{E}_{t,q(Z),\rho_t(X_t|Z)}||v_t(X_t,y) - \dot{X}_t||^2_{g_{X_t}} \tag{7}$$

where $\dot{X} = \frac{d}{dt}X_t = u_t(X_t|Z)$ and $u_t(X_t|Z)$ is the velocity at the geodesic.

**Flow matching on Cholesky manifold.** Recall the definition of *log-Cholesky metric* (eq. 2), we find that many operations on $(\mathcal{S}_n^+, g)$ directly rely on those in $(\mathcal{L}_n^+, \tilde{g})$. Therefore, we seek to optimize the flow on Cholesky manifold directly (Fig. 2 ). Before that, we first need to investigate the feasibility of this approach.

**Lemma 4.1** *[10] The Cholesky map $\mathscr{L} : \mathcal{S}_n^+ \to \mathcal{L}_n^+$ by $\mathscr{L}(LL^\top) = L$ and its inverse $\mathscr{S}$ are isometries between $(\mathcal{S}_n^+, g)$ and $(\mathcal{L}_n^+, \tilde{g})$.*

Based on the *conservation of probability measure* [24] and Lemma 4.1, we have:

**Proposition 4.2** *Given two manifolds $(\mathcal{S}_n^+, g)$ and $(\mathcal{L}_n^+, \tilde{g})$, a probability density $\rho_s$ defined on $(\mathcal{S}_n^+, g)$ together with the isometry $\mathscr{L}$ induces another probability $\rho_l$ on $(\mathcal{L}_n^+, \tilde{g})$ via the relationship:*

$$\rho_s(X) = \rho_l(L), \forall X \in \mathcal{S}_n^+, L = \mathscr{L}(X) \tag{8}$$

The proof of this proposition is given in Appendix. E.1

This proposition enables us to transform our optimization problem from the SPD manifold to the Cholesky manifold while preserving probability measures. Furthermore, we can establish that $q(L_0, L_1) = q(X_0, X_1) = q(Z)$, leading to our modified objective on $\mathcal{L}_n^+$:

$$\mathcal{L}_{\text{RCFM}}(\theta) = \mathbb{E}_{t,q(Z),\rho_t(L_t|Z)}||\tilde{v}_t(L_t,y) - \dot{L}_t||^2_{\tilde{g}_{L_t}} \tag{9}$$

where the velocity $\dot{L}_t$ has a close-form solution derived in Appendix. B.

**Theorem 4.3** *Given two points $X_0, X_1 \in \mathcal{S}_n^+$ and their corresponding Cholesky decomposition factors $L_0, L_1 \in \mathcal{L}_n^+$ where $X_0 = L_0 L_0^T$ and $X_1 = L_1 L_1^T$, let $X_t$ be the geodesic on $\mathcal{S}_n^+$ connecting $X_0$ and $X_1$, and $L_t$ be the corresponding geodesic on $\mathcal{L}_n^+$ connecting $L_0$ and $L_1$. Then at the optimal solution:*

$$\left\| v_t^*(X_t,y) - \dot{X}_t \right\|^2_{g_{X_t}} = \left\| \tilde{v}_t^*(L_t,y) - \dot{L}_t \right\|^2_{\tilde{g}_{L_t}} \tag{10}$$

*where $v_t^*(X_t) = L_t \tilde{v}_t^*(L_t,y)^\top + \tilde{v}_t^*(L_t,y)L_t^\top$ and $\tilde{v}_t^*(L_t)$ is the optimal velocity field on $\mathcal{L}_n^+$. These results demonstrate that minimizing the objective on $\mathcal{L}_n^+$ is equivalent to minimizing the objective on $\mathcal{S}_n^+$.*

The proof of this theorem is given in Appendix. E.2. With these properties, we have the theoretic guarantee to optimize on $\mathcal{L}_n^+$ directly.

This formulation offers several benefits. First, direct optimization on the Cholesky manifold eliminates the need for frequently mapping between $(\mathcal{S}_n^+, g)$ and $(\mathcal{L}_n^+, \tilde{g})$. The more important point is that, this approach avoids the numerical instabilities associated with eigendecomposition in AIRM, particularly for matrices with near-zero eigenvalues.

Algorithm 10 summarizes the training procedure of *BrainFlow*.

### 4.3 Reversing Flows with Consensus Control

Before detailing the mechanism, it is crucial to understand its neuroscientific motivation. The consensus control module is designed to enforce the fundamental biological principle of a shared structural scaffold. The brain's relatively stable anatomical wiring (SC) serves as the foundation for a multitude of dynamic functional states (FCs). Therefore, when reversing the flow from different FCs of the same individual, these pathways should converge towards a single, consistent SC. Our consensus mechanism operationalizes this principle to guide the reverse-flow predictions and ensure they are biologically plausible.

The trained velocity field optimized from eq.9 enables predicting the FCs by simulating $v_t^\theta(\cdot)$ with a given SC. On the flip side, given a functional connectivity, we can get a SC by simulating the reversed vector field $-v_{1-t}^\theta(\cdot)$.

**Proposition 4.4** *Given the Cholesky factor of a functional connectivity $L_1$ of class $y$ and a vector field $\tilde{v}_t^\theta$, by simulating the following ODE from 0 to 1.*

$$\frac{d}{dt}L_t' = -\tilde{v}_{1-t}^\theta(L_t', y), L_0' = L_1, t \in [0, 1] \tag{11}$$

*Then we can get a structural connectivity $L_1'$ corresponding to the FC from same subject if $\tilde{v}_t^\theta(\cdot) = u_t(\cdot|L_1', L_1)$. Here $L_1$ means the result of forward flow (FC), and $L_1'$ means the result of reverse flow (SC).*

However, two factors may cause the simulation to diverge. First, the trained $\tilde{v}_t^\theta$ might not accurately approximate the velocity along the geodesic path. Second, during integration, cumulative errors can cause the trajectory to deviate progressively from the ideal path. Our solution to address these issues is described below.

**Enforces consensus by Dirichlet energy.** We propose a training-free consensus control module based on the idea of consensus in the network system[25]. It aims to minimizing of the Dirichlet energy functional, which enforces consistency among different FC states of the same subject. A fundamental property of the human brain is that its functional processes are supported by its underlying anatomical structure. This implies that FCs across different states share common information rooted in SC. Moreover, SC is relatively stable compared to the varying FCs, making it a shared anchor for all FC states of a subject.

Under this assumption, we define a functional for consistency, $E\left(\{L_1^{y_i}\}_{i=1}^m\right)$, representing the Dirichlet energy of the fully connected network formed by $m$ FC states $\{X_1^{y_i}\}_{i=1}^m$:

$$E\left(\{L_1^{y_i}\}_{i=1}^m\right) = \frac{1}{4}\sum_i^m\sum_{j\neq i}^m a_{ij}\|L_1^{y_j} - L_1^{y_i}\|^2 \tag{12}$$

where each FC factor $L_1^{y_i}$ is regarded as a node of the network and the edge weight between them is defined as $a_{ij} = \exp\left(-\left\|L_1^{y_j} - L_1^{y_i}\right\|^2\right)$, the Gaussian kernel that weights the similarity between the FC states. The Dirichlet energy measures the smoothness of the connectivity states over the fully connected network graph defined by $\{X_1^{y_i}\}$. Intuitively, minimizing $E$ ensures that all FC states become more consistent with each other by reducing their pairwise differences, weighted by their similarity. The negative gradient of $E$ w.r.t. $X_1^{y_i}$ gives the consensus velocity:

$$v_{con}^i = -\nabla_{L_1^{y_i}}E = \sum_{j\neq i}^m a_{ij}\left(L_1^{y_j} - L_1^{y_i}\right) \tag{13}$$

which constructs another flow that as a force pulling $L_1^{y_i}$ to toward its neighbours in the network.

To integrate this consensus mechanism with the original dynamics of the trained velocity field $\tilde{v}_t^\theta$, we modify the ODE governing the evolution of $L_t^{y_i}$ as:

$$\frac{d}{dt}L_t^{y_i} = -\tilde{v}_{1-t}^\theta(L_t, y^i) + \sigma v_{con}^i \tag{14}$$

where $\sigma$ controls the balance between the original dynamics and the consensus term.

After that, we can calculate the Fréchet mean of these $m$ final SC, specifically,

$$L_1' = \frac{1}{m}\sum_{i=1}^m\lfloor L_1'^{y_i}\rfloor + \exp\{m^{-1}\sum_{i=1}^m\log\mathbb{D}(L_1'^{y_i})\} \tag{15}$$

This formulation ensures that the evolution of the FC states is guided by both their individual geodesic paths and a global consensus mechanism rooted in the minimization of Dirichlet energy. We summarize the inference procedure in Algorithm 10.

# 5 Experiments

In this section, we conduct extensive experiments on synthetic datasets and real brain structural-functional network datasets for validating the effectiveness of *BrainFlow*. The detailed implementation and experiment settings are shown in Appendix D and the data description of Human Connectome Project-Aging (HCP-A) [26], HCP-Young Adult (HCP-YA) [27], UK Biobank [28] are listed in Appendix D.4

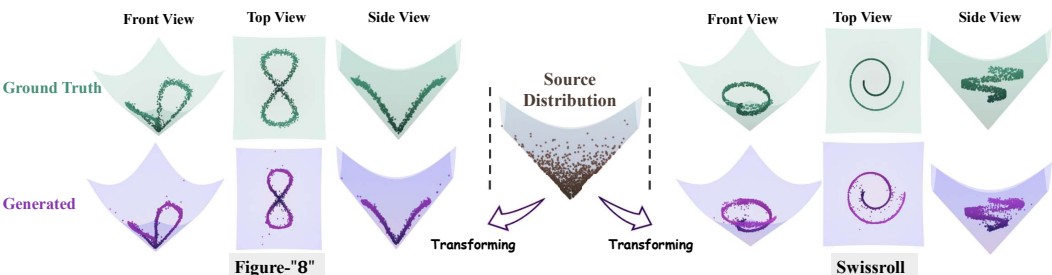

Figure 3: Results of textural distribution by our *BrainFlow*. The top row is the target distribution on SPD manifold. The bottom row shows the distribution transformed from the source distribution. Although the flow operates on the Cholesky manifold, we can accurately recover the results on the SPD manifold, demonstrating the effectiveness of our model.

## 5.1 Experiment on Synthetic Dataset

Fig. 3 and Fig. 5 demonstrate the effectiveness of optimization on the Cholesky manifold and consensus control, respectively, where we demonstrate the promising results of finding a manifold flow from a pre-defined source distribution to the target distribution. Table 7 illustrates the metrics of *BrainFlow* on synthetic dataset. Due to space constraints, further details and analysis are provided in Appendix C.

## 5.2 Functional Connectivity Prediction

*Experimental setting.* We predict functional connectivity from structural connectivity on all three datasets (UK Biobank, HCP-A, HCP-YA). **Baselines** include the one-way connectivity prediction method MGCN-GAN [6] as well as the flow-based method Conditional Flow Matching (CFM)[15], Riemannian Flow Matching on SPD (RCFM) [8], Variance-Preserve Flow Matching (VPFM)[29],Action Matching (AM)[22], Schrödinger Bridge model ([SF]$^2$M) and our *BrainFlow*. To comprehensively evaluate the prediction performance, we employ multiple complementary metrics. The Root-Mean-Square Error (RMSE) quantifies the edge-wise prediction accuracy. To assess the preservation of network topology, we compute the relative error in local clustering coefficients (denoted as LCE, calculated by $\frac{|C_{pred}-C_{real}|}{C_{real}}$), which reflects the accuracy of predicted local network organization around individual nodes, and global clustering coefficient error (denoted as GCE, calculated by $\frac{|C'_{pred}-C'_{real}|}{C'_{real}}$) to evaluate the overall network transitivity prediction. (See Appendix D.2 for details)

*Results.* Table 1 and Fig. 6 present the results of FC prediction. While MGCN-GAN consistently achieves the lowest RMSE across all three datasets, *BrainFlow* excels in preserving network topological properties, as indicated by the lowest GCE and LCE values across all datasets. The SPD variant of RCFM fails to converge across all datasets, as indicated by the divergence notation. This issue arises because, during inference, the intermediate SPD matrix becomes ill-conditioned, with some eigenvalues approaching zero. Moreover, the training time per epoch is significantly longer, taking approximately **100** seconds under the experimental setting described in Appendix D, whereas *BrainFlow* requires only **8** seconds per epoch.

*Discussion.* This performance trade-off between edge-wise accuracy (RMSE) and topological preservation (GCE, LCE) suggests that different models prioritize distinct aspects of the prediction task. While MGCN-GAN excels at minimizing direct edge-wise errors, *BrainFlow* more effectively preserves the network's hierarchical organization at both local and global scales. The visualization

in Fig. 6 highlights that our method better preserves the contrast between different connectivity strengths, maintaining a clearer separation between strongly connected regions (yellow) and weakly connected areas (blue)—a crucial factor for accurately representing brain network organization.

Table 1: Performance comparison across different datasets. **Bold** is the best.

| Model | UK Biobank | | | HCP-A | | | HCP-YA | | |
|---|---|---|---|---|---|---|---|---|---|
| | RMSE | LCE (%) | GCE (%) | RMSE | LCE (%) | GCE (%) | RMSE | LCE (%) | GCE (%) |
| MGCN-GAN | **0.219±0.033** | 19.0±1.4 | 10.9±2.6 | **0.217±0.038** | 15.7±1.4 | 19.0±3.8 | **0.243±0.033** | 14.0±1.4 | 23.7±2.5 |
| CFM | 0.266±0.004 | 9.5±0.4 | 6.1±0.6 | 0.259±0.003 | 11.4±0.3 | 8.3±0.5 | 0.280±0.002 | 13.3±0.3 | 9.4±0.8 |
| RCFM (SPD) | diverge | diverge | diverge | diverge | diverge | diverge | diverge | diverge | diverge |
| AM | 0.982±0.001 | 47.0±0.2 | 46.6±0.2 | 0.962±0.001 | 48.6±0.2 | 48.2±0.2 | 0.971±0.002 | 49.7±0.2 | 49.1±0.2 |
| VPFM | 0.274±0.001 | 10.4±0.3 | 9.9±0.7 | 0.445±0.211 | 17.2±13.3 | 19.1±12.2 | 0.304±0.002 | 10.4±0.1 | 8.8±1.8 |
| $[SF]^2M$ | 0.293±0.006 | 10.0±1.6 | 6.6±1.6 | 0.267±0.002 | 9.1±0.3 | 5.9±0.8 | 0.355±0.001 | 9.9±0.4 | 7.4±1.0 |
| *BrainFlow* | 0.290±0.001 | **8.9±0.3** | **5.5±0.3** | 0.291±0.002 | **8.1±0.2** | **4.5±0.2** | 0.336±0.001 | **8.7±0.2** | **5.8±0.5** |

Table 2: Generated Samples Classification Performance Across Datasets (N/A means the metrics fall below a meaningful threshold for application).

| Model | UK Biobank (Binary) | | HCP-A (4 tasks) | | HCP-YA (7 tasks) | |
|---|---|---|---|---|---|---|
| | F1 (%) | ACC (%) | F1 (%) | ACC (%) | F1 (%) | ACC (%) |
| Real | 99.12 | 99.10 | 95.51 | 93.48 | 92.31 | 92.06 |
| AM | N/A | N/A | N/A | N/A | N/A | N/A |
| VPFM | 98.44±1.04 | 98.28±1.22 | 44.60±34.79 | 42.22±34.48 | 56.64±1.77 | 60.16±2.16 |
| $[SF]^2M$ | 98.65±0.26 | 98.67±0.26 | 77.05±2.07 | 82.99±1.46 | 51.36±3.94 | 53.87±3.80 |
| CFM | 98.59±0.30 | 98.59±0.28 | 77.03±2.12 | **85.48±1.85** | 57.89±3.55 | 59.87±3.14 |
| *BrainFlow* | **98.85±0.17** | **98.93±0.19** | **77.17±2.34** | 83.40±2.51 | **61.14±4.67** | **64.23±4.49** |

Table 3: Reverse Flow Performance Evaluation Across Different Datasets.

| Reverse Flow | UK Biobank | | | HCP-A | | | HCP-YA | | |
|---|---|---|---|---|---|---|---|---|---|
| | RMSE | LCE (%) | GCE (%) | RMSE | LCE (%) | GCE (%) | RMSE | LCE (%) | GCE (%) |
| AM | 1.17±0.00 | 0.24±0.02 | 0.21±0.03 | 1.50±0.00 | 26.23±1.80 | 26.70±1.90 | 1.46±0.00 | 25.51±1.70 | 25.54±1.81 |
| VPFM | 0.27±0.01 | 0.18±0.01 | 0.18±0.01 | 0.55±0.59 | 10.71±12.41 | 11.72±13.65 | 0.31±0.00 | 0.55±0.04 | 0.57±0.04 |
| $[SF]^2M$ | 0.23±0.00 | 0.20±0.01 | 0.21±0.00 | 0.25±0.00 | **0.48±0.06** | **0.49±0.05** | 0.30±0.00 | 0.51±0.02 | 0.51±0.02 |
| CFM | 1.14±0.00 | 24.63±0.02 | 26.15±0.02 | 1.03±0.00 | 20.23±0.04 | 21.46±0.04 | 0.92±0.00 | 21.28±0.01 | 23.01±0.01 |
| *BrainFlow* | **0.07±0.00** | **0.09±0.00** | **0.08±0.00** | 0.36±0.00 | 0.59±0.03 | 0.59±0.04 | **0.24±0.00** | **0.47±0.02** | **0.46±0.02** |

Table 4: Ablation study on noise level $\sigma$ across different datasets. **Bold** indicates the best performance for each metric.

| Weight Parameter $\sigma$ | HCP-A | | | HCP-YA | | | UK Biobank | | |
|---|---|---|---|---|---|---|---|---|---|
| | MSE | LCE (%) | GCE (%) | MSE | LCE (%) | GCE (%) | MSE | LCE (%) | GCE (%) |
| 0.0 | 0.65±0.00 | 0.74±0.00 | 0.73±0.00 | 0.42±0.00 | 0.62±0.00 | 0.59±0.00 | 0.09±0.00 | 0.52±0.00 | 0.53±0.00 |
| 0.2 | **0.13±0.00** | **0.71±0.00** | **0.69±0.00** | **0.14±0.00** | **0.55±0.00** | **0.59±0.00** | **0.07±0.00** | **0.39±0.00** | **0.39±0.00** |
| 0.4 | 0.44±0.01 | 0.95±0.00 | 0.93±0.01 | 0.52±0.00 | 0.93±0.00 | 0.88±0.00 | 0.69±0.00 | 1.33±0.01 | 1.30±0.01 |
| 0.6 | 1.52±0.03 | 3.17±0.03 | 3.18±0.02 | 1.32±0.02 | 2.92±0.01 | 3.02±0.02 | 1.89±0.002 | 10.93±0.02 | 11.16±0.01 |
| 0.8 | 2.37±0.01 | 20.28±0.05 | 20.50±0.03 | 2.12±0.01 | 15.74±0.03 | 16.40±0.02 | 2.52±0.01 | 27.88±0.02 | 28.16±0.02 |

## 5.3 Brain Task Recognition Using The Predicted FC

*Experimental description.* In this experiment, we evaluate the conditional prediction ability of *BrainFlow*. However, the differences in connectivity patterns across cognitive tasks are often highly variable across subjects, making it challenging to establish consistent signatures for specific task states. Additionally, no standardized metric exists to evaluate the quality of task-specific connectivity predictions. In this context, we employ proxy evaluation metrics such as accuracy and F1-score within a downstream task-classification framework. Specifically, we first train a classifier on the real FC data. We then apply this pre-trained classifier to the generated FC to assess whether they retain sufficient task-specific information for accurate task discrimination. Additionally, we generate samples for each task in the HCP-YA dataset and then average them within each task to obtain a representative sample each task. We then compare them with the real average representation. Appendix D.5 shows related result.

*Results.* Table 2 illustrates the effectiveness of *BrainFlow* in preserving task-specific information across datasets with varying task complexity. For the UK Biobank dataset, which involves binary

task classification, the model achieves an impressive 98.93% accuracy with the generated functional connectivity, closely matching the real data performance of 99.12%. This result demonstrates that *BrainFlow* effectively captures the critical connectivity patterns necessary for distinguishing between the two cognitive states in binary task discrimination. As task complexity increases, prediction performance declines but remains superior to the other methods in most scenarios.

## 5.4 Reverse Structural Connectivity Prediction

*Experimental description.* In this experiment, we evaluate the performance of consensus control by reversing the trained *BrainFlow*, using different FC inputs (We have multiple fMRI scans for each subject) from the same subject. We employ the same evaluation metrics as in Sec. 5.2, including RMSE, LCE, and GCE.

*Results.* Table 3 highlights the effectiveness of our proposed consensus control mechanism across three datasets. Notably, most approaches achieve consistently low error rates, largely due to the inherent stability of SC patterns across subjects, in contrast to the more dynamic nature of FC. This fundamental characteristic of brain architecture ensures that the core graph topology remains well-preserved during reconstruction, as reflected by the consistently low clustering coefficient errors observed across all datasets. For a visual comparison, please refer to Fig. 7 in Appendix D.5.

*Discussion.* The most notable improvements are observed in the UK Biobank dataset, where the consensus control mechanism significantly reduces both global and local clustering coefficient errors while maintaining a low RMSE. This indicates that the consensus approach effectively integrates shared information across multiple FC states to reconstruct more accurate structural patterns.

## 5.5 Parameter Sensitivity Analysis

*Experimental description.* We conduct an experiment to test the sensitivity of the consensus control parameter $\sigma$. We evaluate BrainFlow's reverse flow performance across five different $\sigma$ values: 0.0, 0.2, 0.4, 0.6, 0.8 on all three datasets (UK Biobank, HCP-A, HCP-YA). For each configuration, we use the same evaluation metrics as in Sec. 5.2, including MSE, LCE, and GCE. The $\sigma = 0.0$ setting represents vanilla reverse flow without consensus control, serving as our baseline for comparison.

*Results.* Table 4 presents the results across different $\sigma$ values. The optimal performance is consistently achieved at $\sigma = 0.2$ across all three datasets, with substantially lower error metrics compared to $\sigma = 0.0$ (vanilla reverse flow).

*Discussion.* We can see that when $\sigma$ is small ($\sim 0.2$), the model has significant performance gain compared to vanilla reverse flow. However, as $\sigma$ increases, the consensus control comes to dominate the reverse flow, causing the trajectories to become biased. This degradation occurs because excessive consensus control overly constrains the reverse trajectories, causing them to converge prematurely and lose fidelity to individual structural patterns. We then argue that $\sigma$ may not be suitable to optimize during training. First, the reversible nature of our flow means that Dirichlet energy regularization applied to the reverse flow also constrains the forward flow, thereby reducing the diversity of predicted task FCs. Second, there is a mismatch between training and inference objectives. While consensus control leverages complementary information across different task FCs to reduce simulation error during inference, BrainFlow training is simulation-free and does not account for this simulation error. Therefore, optimizing $\sigma$ during training is unsuitable for the intended inference procedure.

# 6 Conclusion

In this work, we propose a flow matching approach on the Cholesky manifold to bridge SC and FC, improving numerical stability and efficiency. A consensus control mechanism enforces consistency across functional states, mitigating errors from vector field approximations and numerical integration. Our results demonstrate effective structural connectivity inference, with applications in neuroscience, brain disorder diagnosis, and connectivity-based interventions. Future work includes extending to dynamic functional connectivity modeling and integrating domain-specific constraints. It could be extended to model both macro-scale lifespan connectivity changes and micro-scale, within-scan fluctuations by learning flows between consecutive time windows.

## Acknowledgement

This work was supported by the National Institutes of Health (AG091653, AG068399, AG084375) and the Foundation of Hope. The views and conclusions contained in this document are those of the authors and should not be interpreted as representing the official policies, either expressed or implied, of the NIH.

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

# A   Additional Basic of Log-Cholesky Metric

We follow the notations on [10]. Denote the Cholesky decomposition of $P \in \mathcal{S}_n^+$ as $P = LL^\top$, the Cholesky map is defined as $\mathscr{L}(P) = L$ and its inverse map is $\mathscr{S}(L) = P$. Table. 5 shows properties of Riemannian manifolds $(\mathcal{L}_n^+, \tilde{g})$ and $(S_n^+, g)$.

Table 5: Basic properties of Riemannian manifolds.

| Basic properties of Riemannian manifolds $(\mathcal{L}_n^+, \tilde{g})$ and $(S_n^+, g)$ | |
| --- | --- |
| $(\mathcal{L}_n^+, \tilde{g})$ | $(S_n^+, g)$ |
| tangent space at $L$ | tangent space at $P$ |
| $\mathcal{L}_n$ | $\mathcal{S}_n$ |
| Riemannian metric | Riemannian metric |
| $\tilde{g}_L(X, Y) = \sum_{i>j} X_{ij} Y_{ij} + \sum_{j=1}^n X_{jj} Y_{jj} L_{jj}^{-2}$ | $g_P(W, V) = \tilde{g}_{\mathscr{L}(P)}((D_P \mathscr{L})(W), (D_P \mathscr{L})(W))$ |
| geodesic emanating from $L$ with direction $X$ | geodesic emanating from $P$ with direction $W$ |
| $\tilde{\gamma}_{L,X}(t) = \lfloor L \rfloor + t\lfloor X \rfloor + \mathbb{D}(L) \exp\{t\mathbb{D}(X)\mathbb{D}(L)^{-1}\}$ | $\gamma_{P,W}(t) = \tilde{\gamma}_{\mathscr{L}(P),(D_P\mathscr{L})(W)}(t) \tilde{\gamma}_{\mathscr{L}(P),(D_P\mathscr{L})(W)}(t)^\top$ |
| Riemannian exponential map at $L$ | Riemannian exponential map at $P$ |
| $\widetilde{\mathrm{Exp}}_L X = \lfloor L \rfloor + \lfloor X \rfloor + \mathbb{D}(L) \exp\{\mathbb{D}(X)\mathbb{D}(L)^{-1}\}$ | $\mathrm{Exp}_P W = \widetilde{\mathrm{Exp}}_{\mathscr{L}(P)}(D_P\mathscr{L})(W)\{\widetilde{\mathrm{Exp}}_{\mathscr{L}(P)}(D_P\mathscr{L})(W)\}^\top$ |
| Riemannian logarithmic map at $L$ | Riemannian logarithmic map at $P$ |
| $\widetilde{\mathrm{Log}}_L K = \lfloor K \rfloor - \lfloor L \rfloor + \mathbb{D}(L) \log\{\mathbb{D}(L)^{-1}\mathbb{D}(K)\}$ | $\mathrm{Log}_P Q = (D_{\mathscr{L}(P)}\mathscr{L})(\widetilde{\mathrm{Log}}_{\mathscr{L}(P)}\mathscr{L}(Q))$ |
| geodesic distance between $L$ and $K$ | geodesic distance between $P$ and $Q$ |
| $d_{\mathcal{L}_n^+}(L, K) = \{\|\lfloor L \rfloor - \lfloor K \rfloor\|_F^2 + \|\log \mathbb{D}(L) - \log \mathbb{D}(K)\|_F^2\}^{1/2}$ | $d_{S_n^+}(P, Q) = d_{\mathcal{L}_+}(\mathscr{L}(P), \mathscr{L}(Q))$ |

# B   Extensions of methods

## B.1   Solution for closed-form velocity.

Given the flow/geodesic $L_t$, the constant velocity $\dot{L}_t$ in eq.9 is calculated by $\frac{d}{dt}L_t = \dot{L}_t$ [8], meaning that the conditional velocity field $u_t(L_t|Z)$ can be obtained by calculating the geodesic and its derivative *w.r.t.* time. While this could be calculated using autograd during the forward pass, we derive a more efficient closed-form expression based on the geodesic definition (eq. 3) on $(\mathcal{L}_n^+, \tilde{g})$:

$$\dot{L}_t = \dot{\tilde{\gamma}}_{L_0,L_1}(t) = \lfloor L_0 \rfloor - \lfloor L_1 \rfloor + \mathbb{D}(L)\exp(tM)M$$
$$M = \log\mathbb{D}(L_0) - \log\mathbb{D}(L_1) \tag{16}$$

## B.2   Computation Cost Analysis

We provide a theoretical analysis comparing the computational complexity of *BrainFlow* against **RCFM-SPD** and vanilla Flow Matching. Flow-based methods can be summarized with the following pipeline steps:

- Sampling from the source and target distributions
- Calculate the interpolated position and velocity
- Regress the predicted velocity to ground truth velocity
- During inference, integrate the time-dependent velocity

We analyze the additional computation of Riemannian conditional flow matching on SPD manifold (**RCFM-SPD**) and *BrainFlow* on these steps compared to baseline vanilla Flow Matching.

**Sampling Stage.** Before sampling stage, additional preprocessing (Cholesky decomposition) is required for BrainFlow, whose time complexity is $\mathcal{O}(Nd^3)$, where $N$ is the size of training set and $d$ is the matrix dimension. Importantly, the computational cost of Cholesky preprocessing is incurred **only once**, whereas the costs associated with **interpolation, loss calculation, and inference** need to be **repeated at each step**.

**Interpolation Stage.** During interpolation stage, the $\exp(\cdot), \log(\cdot)$ operations of **RCFM-SPD** both require $\mathcal{O}(Bd^3)$, where $B$ is the batch size, while these operations on Cholesky manifold only

requires $\mathcal{O}(Bd)$, as the strict lower triangular part follows element-wise addition/subtraction and the diagonal part uses diagonal matrix exponential and logarithm.

**Loss Calculation Stage.** During loss calculation, the inner product $\langle \cdot, \cdot \rangle$ on SPD manifold requires $\mathcal{O}(Bd^3)$ while Cholesky manifold has the same complexity as vanilla Flow Matching ($L_2$ norms).

**Inference Stage.** During inference, **RCFM-SPD** requires eigenvalue clipping to keep the element SPD, requiring $\mathcal{O}(Bd^3)$ each integration step. Hence, it requires additional $\mathcal{O}(LBd^3)$ where $L$ is total step, while BrainFlow naturally maintains the Cholesky structure without additional operations.

Table 6 summarizes the computational complexity across different stages. The key insight is that while BrainFlow incurs a one-time preprocessing cost, RCFM-SPD repeatedly pays $\mathcal{O}(Bd^3)$ at every iteration and $\mathcal{O}(LBd^3)$ during inference, making BrainFlow substantially more efficient in practice.

Table 6: Computational complexity comparison. N: training set size, B: batch size, d: matrix dimension, L: integration steps. "+" denotes additional computational cost compared to baseline.

| Stage | RCFM-SPD | BrainFlow | Flow Matching |
|---|---|---|---|
| Sampling | Standard | $+\mathcal{O}(Nd^3)$ Cholesky preprocessing | Baseline |
| Interpolation | $+\mathcal{O}(Bd^3) \exp(\cdot), \log(\cdot)$ **(SPD)** | $+\mathcal{O}(Bd) \exp(\cdot), \log(\cdot)$ **(Cholesky)** | Baseline |
| Loss | $+\mathcal{O}(Bd^3)$ **(SPD inner product)** | Standard **(Cholesky inner product)** | Baseline |
| Inference | $+\mathcal{O}(LBd^3)$ eigenvalue clipping | Standard | Baseline |

For typical brain connectivity matrices with $d = 116$ regions and $L = 100$ integration steps, the repeated eigendecompositions in RCFM-SPD accumulate to significantly higher computational costs than BrainFlow's one-time preprocessing. Empirically, as reported in Sec. 5.2, BrainFlow achieves a **12.5×** speedup (8 seconds vs. 100 seconds per epoch) while maintaining numerical stability throughout training and inference.

### B.3 Algorithm

Here we briefly introduce the algorithm pipeline in Alg. 10

---

**Algorithm 1** Training of *BrainFlow*

**Require:** Joint distribution $q(X_0, X_1)$
  Initialize parameters $\theta$ of $\tilde{v}_t^\theta$
  **while** not converged **do**
    sample time $t \sim \text{uniform}(0, 1)$
    sample training pair $(X_0, X_1) \sim q$ with task label $y$
    $L_0, L_1 = \mathscr{L}(X_0), \mathscr{L}(X_1)$
    $L_t = \exp_{L_0}(t \log_{L_0}(L_1))$
    calculate $\dot{L}_t$ from eq. 16
    $\mathcal{L}(\theta) = \|\tilde{v}_t^\theta(L_t, y) - \dot{L}_t\|_{\tilde{g}_{L_t}}^2$
    $\theta = \text{optimizer\_step}(\mathcal{L}(\theta))$
  **end while**

---

**Algorithm 2** Predict FC/SC

**Require:** a SC $X_0$ with desired task state $y$ or $m$ FCs from same subject $\{X_1^{y_i}\}_{i=1}^m$, trained $v^\theta$
  **if** SC (forward flow) **then**
    $L_0 = \mathscr{L}(X_0)$
    $L_1 = \text{ode\_solve}(L_0, v_t^\theta(\cdot, y), t = (0, 1))$
    $X_1 = L_1 L_1^\top$
  **else if** FC (reverse flow) **then**
    $L_1^{y_i} = \mathscr{L}(X_1^{y_i}), i = 1, .., m$
    $L_1^{'y_i} = \text{ode\_solve}(L_1^{y_i}, -v_{1-t}^\theta(\cdot, y_i) + \sigma v_{con}^i, t = (0, 1))$
    calculate $L1'$ using eq.15 with $\{L_1^{'y_i}\}_{i=1}^m$
    $X_1' = L_1' L_1'^\top$
  **end if**
  **return** Predicted FC $X_1$ or SC $X_1'$

# C  Experiment on Synthetic Dataset

## C.1  Generation of Artificial Textures on SPD Manifold

We trained a simple MLP-based Flow Matching (FM) model on Cholesky manifold. Two synthetic datasets whose elements are SPD matrices on $\mathcal{S}_2^+$ are used to visualize the trained dynamics. They are the *siwssroll* and *figure-eight*. We can represent the space of $2 \times 2$ SPD matrices $\mathcal{S}_2^+$ as points above a circular cone in 3D space by mapping each SPD matrix to a point in $\mathbb{R}^3$. The cone itself represents singular matrices with zero determinant. Every point above this cone corresponds to exactly one SPD matrix for visualization.

**Result.** We apply Cholesky decomposition to both the prior and target dataset and transformed them back to SPD manifold. The results of data generation are shown in Fig. 3. Fig. 3 presents three orthogonal views (Front, Top, and Side) of both a *Figure-"8"* pattern and a *Swissroll* configuration. For each pattern, we compare the original data distribution (shown in green) with the samples generated by the FM on Cholesky manifold (shown in purple). All generated points strictly remain within the SPD manifold, as evidenced by their positioning above the determinant-zero cone in all views. The visual consistency between the generated and original samples across both patterns, coupled with the strict adherence to the SPD manifold constraints, demonstrates that learning flow on the Cholesky manifold is practical.

## C.2  Quantity Performance on Synthetic Datasets

**Dataset** Besides the low-dimensional artificial textures data on SPD manifold, we also generate two $100 \times 100$ SPD matrix distributions for further analysis. Specifically, we first generate two groups of random matrices with different mean values and standard variance. Then we can get SPD matrices by $X = LL^\top + \epsilon I$ where $L$ is the random matrix and $I$ is the identity matrix. We evaluate the Wasserstein distance between the generated and true distributions. The ratio of generated samples that lie on SPD manifold is also considered.

**Result** Table 7 shows that our method can keep the SPD property well and get lower Wasserstein distance.

Table 7: Model Performance across Synthetic Datasets. ($W_2$) means the quadratic Wasserstein distance

| Model | Swiss-roll | | Fig-8 | | High-dim SPD | |
|---|---|---|---|---|---|---|
| | SPD ratio (%) | $W_2(\times 10^{-4})$ | SPD ratio (%) | $W_2(\times 10^{-3})$ | SPD ratio (%) | $W_2(\times 10^{-2})$ |
| AM | 100.00 | 9.03 | 100.00 | 1.42 | 0.00 | 9.13 |
| VPFM | 99.96 | 6.28 | 99.99 | 1.93 | 0.00 | 7.98 |
| [SF]$^2$M | 99.98 | 6.05 | 99.69 | 1.81 | 71.98 | 7.98 |
| CFM | 98.84 | 5.98 | 99.95 | 1.83 | 23.78 | 7.98 |
| RCFM | 100.00 | 4.02 | 100.00 | 1.43 | 100.00 | 8.00 |
| *BrainFlow* | 100.00 | 4.07 | 100.00 | 1.43 | 100.00 | 7.97 |

## C.3  Controlled Reverse Flow on Synthetic Datasets

In this experiment we validate the reverse ODE flow with the proposed consensus control. For simplicity, we use synthetic datasets and FM model on Euclidean space to illustrate the effect of consensus control.

**Dataset.** The synthetic dataset consists of points in 2D Euclidean space constructed as follows. The source distribution contains points uniformly distributed on a circle with radius $0.5$, with small random perturbations ($\sigma = 0.05$) added to create a natural distribution (Green Circle in Fig. 5). Each

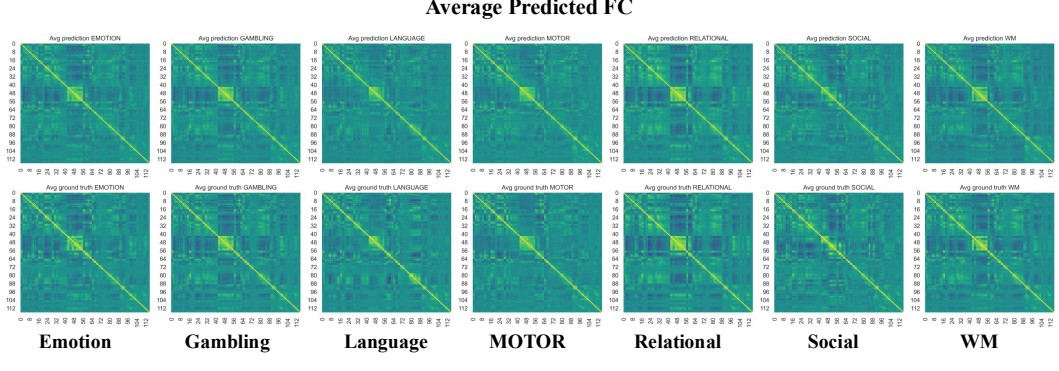

| Emotion | Gambling | Language | MOTOR | Relational | Social | WM |

**Average True FC**

Figure 4: The results of average predicted task FC and average true task FC in HCP-YA dataset

point in the source distribution is assigned a unique identifier (ID). These points are then transformed into the target distribution through two distinct mappings corresponding to different classes. Points mapped to first class are expanded to a circle with radius $1.0$ and rotated clockwise by $\frac{\pi}{6}$ (Orange circle in Fig. 5), while points mapped to the second class are contracted to another larger circle with radius $1.5$ and rotated counter-clockwise by $\frac{\pi}{6}$ (Blue circle in 5). Both transformations include small Gaussian noise ($\sigma = 0.05$) to simulate real-world variations. Finally, points in target distribution inherit their IDs from their source points, meaning that points from different classes can share the same ID if they originated from the same source point. This design allows us to clearly demonstrate how consensus control can help points with the same ID converge to their true source, even when the conditional optimal transport coupling suggests different correspondences.

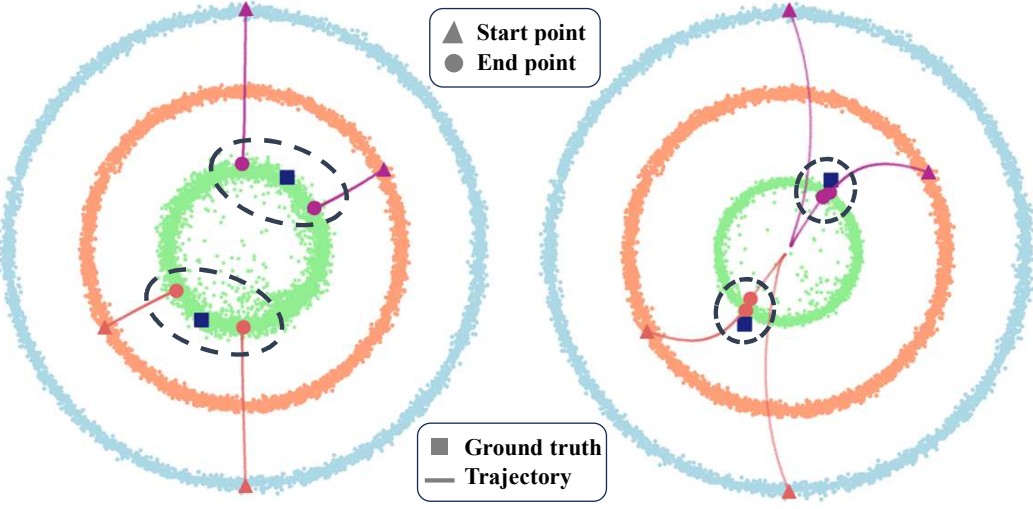

**Trajectory without Control**     **Trajectory with Control**

Figure 5: Examples of consensus control. The left panel illustrates reverse flow without control, deviating significantly from the ground truth. The right panel demonstrates reverse flow with control, effectively steering it closer to the desired outcome.

**Result.** The FM model is trained with conditional OT path and we sample points independently from source distribution and target distributions. Fig. 5 (left) shows the reversed trained flow from the target distribution to the source distribution and the right one shows the reversed flow with consensus control. Two groups of sample points and their trajectories are shown in the Fig. 5 with different colours (purple and orange).

In the case without control (left), the trajectories of points sharing the same ID (highlighted by blue dashed circles) fail to converge to their true source positions. The flow merely follows the learned conditional optimal transport paths, leading to divergent endpoints even for points that originated

Table 8: Task-specific functional connectivity prediction performance on HCP-YA dataset. Best results in **bold**.

| Task | Metric | AM | VPFM | $[SF]^2M$ | CFM | BrainFlow |
|------|--------|-----|-------|-----------|-----|-----------|
| Emotion | RMSE | 0.980±0.002 | 0.339±0.003 | 0.367±0.002 | 0.314±0.001 | 0.329±0.002 |
|  | LCE (%) | 0.503±0.002 | 0.089±0.005 | 0.093±0.005 | 0.138±0.002 | **0.085±0.001** |
|  | GCE (%) | 0.499±0.003 | 0.080±0.009 | 0.064±0.010 | 0.095±0.004 | **0.059±0.003** |
| Gambling | RMSE | 0.974±0.002 | 0.313±0.002 | 0.343±0.003 | 0.288±0.002 | 0.303±0.002 |
|  | LCE (%) | 0.481±0.003 | 0.100±0.010 | 0.104±0.008 | 0.125±0.001 | **0.097±0.002** |
|  | GCE (%) | 0.473±0.003 | 0.060±0.022 | 0.076±0.017 | 0.072±0.002 | **0.071±0.003** |
| Language | RMSE | 0.965±0.001 | 0.287±0.002 | 0.313±0.003 | 0.268±0.001 | 0.282±0.002 |
|  | LCE (%) | 0.526±0.002 | 0.090±0.006 | 0.097±0.007 | 0.119±0.002 | **0.084±0.001** |
|  | GCE (%) | 0.521±0.002 | 0.057±0.013 | 0.077±0.016 | 0.083±0.002 | **0.059±0.001** |
| Motor | RMSE | 0.969±0.003 | 0.302±0.003 | 0.325±0.002 | 0.280±0.002 | 0.295±0.002 |
|  | LCE (%) | 0.526±0.005 | 0.089±0.008 | 0.085±0.004 | 0.117±0.001 | **0.081±0.001** |
|  | GCE (%) | 0.524±0.005 | 0.083±0.016 | 0.053±0.011 | 0.083±0.001 | **0.057±0.002** |
| Relational | RMSE | 0.976±0.002 | 0.317±0.003 | 0.352±0.003 | 0.293±0.001 | 0.307±0.003 |
|  | LCE (%) | 0.453±0.005 | 0.106±0.010 | 0.112±0.011 | 0.128±0.002 | **0.101±0.002** |
|  | GCE (%) | 0.442±0.005 | 0.071±0.022 | 0.084±0.026 | 0.096±0.002 | **0.070±0.003** |
| Social | RMSE | 0.971±0.002 | 0.303±0.003 | 0.330±0.003 | 0.281±0.002 | 0.295±0.002 |
|  | LCE (%) | 0.491±0.003 | 0.095±0.009 | 0.102±0.006 | 0.133±0.001 | **0.090±0.001** |
|  | GCE (%) | 0.482±0.003 | 0.063±0.021 | 0.084±0.013 | 0.064±0.002 | **0.066±0.003** |
| WM | RMSE | 0.961±0.001 | 0.269±0.003 | 0.298±0.003 | 0.249±0.002 | 0.262±0.002 |
|  | LCE (%) | 0.502±0.006 | 0.095±0.007 | 0.101±0.012 | 0.125±0.002 | **0.090±0.003** |
|  | GCE (%) | 0.493±0.007 | 0.089±0.012 | 0.076±0.018 | 0.075±0.002 | **0.059±0.003** |

from the same source. In contrast, when consensus control is applied (right), the trajectories of points with shared IDs successfully converge to their corresponding ground truth source positions, as indicated by their alignment with the square markers representing ground truth locations. This convergence is achieved despite the points starting from different target distributions (orange and blue circles) and having different initial flow directions. The controlled trajectories exhibit smooth adaptation from their initial paths toward consensus points, demonstrating the effectiveness of the control mechanism in enforcing consistent reverse mapping.

# D    Experiment Settings

## D.1    Model Implementation

We base our implementation on [30]. The flow-based approach uses an 8-layer Transformer [31] without positional encoding to maintain permutation equivariance of connectivity matrices. We use

a label embedding to handle conditional predictions, while time information follows the encoding scheme from [30]. For inference, we employ an Euler solver with a 0.01 step size.

## D.2 Clustering Coefficient Calculation

We provide detailed explanations for the clustering coefficient metrics used throughout this paper.

**Graph Notation.** An undirected graph $G = (V, E)$ formally consists of a set of vertices $V$ and a set of edges $E$ between them. An edge $e_{ij}$ connects vertex $v_i$ with vertex $v_j$. For functional connectivity matrices, we only consider positive edge weights.

**Local Clustering Coefficient (LCC).** The local clustering coefficient for undirected graphs quantifies the degree to which neighbors of a given node are connected to each other, defined as:

$$C_i = \frac{2|e_{jk} : v_j, v_k \in N_i, e_{jk} \in E|}{k_i(k_i - 1)} \tag{17}$$

where $N_i$ denotes the neighborhood of node $i$, and $k_i = \sum_j A_{ij}$ is the degree of node $i$.

Expressed in terms of the adjacency matrix $A$, this becomes:

$$C_i = \frac{1}{k_i(k_i - 1)} \sum_{j,k} A_{ij} A_{jk} A_{ki} \tag{18}$$

which counts the number of closed triangles (triplets) involving node $i$.

**Global Clustering Coefficient (GCC).** The global clustering coefficient measures the overall tendency of nodes to form clusters across the entire network:

$$C = \frac{\text{number of closed triplets}}{\text{number of all triplets (open and closed)}} \tag{19}$$

In matrix form:

$$C = \frac{\sum_{i,j,k} A_{ij} A_{jk} A_{ki}}{\frac{1}{2} \sum_i k_i(k_i - 1)} \tag{20}$$

where $C = 0$ when the denominator is zero (i.e., no connected triplets exist).

**Error Metrics.** We compute the relative error for both local and global clustering coefficients as:

$$\text{LCE} = \frac{|C_{\text{pred}} - C_{\text{real}}|}{C_{\text{real}}} \tag{21}$$

$$\text{GCE} = \frac{|C'_{\text{pred}} - C'_{\text{real}}|}{C'_{\text{real}}} \tag{22}$$

where $C$ denotes local clustering coefficient and $C'$ denotes global clustering coefficient. For isolated nodes where $C_{\text{real},i} = 0$, we exclude these nodes from the LCE calculation as they represent degenerate cases and occur very rarely in brain connectivity networks.

## D.3 Training Details

The datasets are split into train, validation, test set with ratio 7:1:2. All the models are trained with 1000 epoch using AdamW [32] with Cosine Annealing learning rate schedule [33]. We do grid search on the following hyperparameters: (1) learning rate: $\{0.001, \mathbf{0.0005}, 0.0001\}$ (2) Layer of Model: $\{4, 6, \mathbf{8}, 12\}$ and (3) Batch size: $\{256, 512, \mathbf{1024}\}$. We do model evaluation every 50 epochs. All the experiments are repeated 5 times with different random seed for data split and model initialization. The structural connectivity data is preprocessed by first $A^s = \log(A^s + 1)$, where the $\log(\cdot)$ is element-wise logarithm. This step can constrain the scale of SC to a reasonable range. Then we add a large enough $\epsilon I$ to the SC, transforming it to SPD matrix. We run all the experiments using a single NVIDIA A6000 GPU.

## D.4 Data Description

**Data Processing** For functional connectivity preprocessing and construction, we use xcp-d [34], a post-processing pipeline for fMRI data. As for the structural connectivity, we utilize QSIPrep [35] for processing and reconstruction. XCP-D computes functional connectivity matrices using Pearson correlation coefficients between parcellated time series. The pipeline employs the 36P nuisance regression strategy, which includes six motion parameters along with their derivatives and quadratic terms, plus white matter, cerebrospinal fluid, and global signal regressors. The correlation calculations themselves do not include any explicit regularization beyond the standard nuisance regression approach. QSIPrep's default reconstruction workflows use iFOD2 (2nd-order Integration over Fiber Orientation Distributions) as the tractography algorithm, and SIFT2 (Spherical-deconvolution Informed Filtering of Tractograms 2) is enabled to provide biologically meaningful connection weights.

**HCP-A** Human Connectome Project-Aging (HCP-A) dataset includes data from 717 subjects, encompassing both fMRI (4,846 time series) and Diffusion Weighted Imaging (DWI) (717) scans [26]. This rich collection facilitates in-depth analyses of both functional and structural connectivity. The HCP-A dataset includes data from four brain tasks associated with memory: VISMOTOR, CARIT, FACENAME, and resting state. In the following experiments, these tasks are treated as distinct categories. In the following experiments, we treat the data as a four-class classification problem.

**HCP-YA** Human Connectome Project - Young Adults (HCP-YA) database [27] involves 11,608 fMRI and 328 DWI. Each fMRI scan includes seven cognitive tasks associated with memory, including Motor, Relational, Social, Working memory, Language, Emotion, and Gambling. In the following experiments, we treat the data as a seven-class classification problem.

**UK Biobank** is a large-scale neuroimaging dataset that includes MRI data [28]. Specifically, it contains preprocessed fMRI ($n$=14,619) and DWI ($n$=5,731 data, following the pipeline described in [36]. The dataset includes recordings from a brain task designed to engage both cognitive and sensorimotor functions [37]. In the following experiments, we treat the data as a two-class classification problem.

For all these three datasets, we partition each SC and FC into 116 regions using AAL atlas [38]. Thus, SC is a $116 \times 116$ matrix where each element is quantified by the number of nerve fibers linking two brain regions.

## D.5 Further Results

**SC-to-FC.** The visualization at Fig. 6 reveals that our method better preserves the contrast between different connectivity strengths, maintaining a more distinct separation between strongly connected regions (yellow) and weakly connected areas (blue), which is crucial for accurate representation of brain network organization. This qualitative observation reinforces our quantitative findings that *BrainFlow* excels in preserving the hierarchical organization of brain networks, even though it may not minimize edge-wise errors to the same degree as MGCN-GAN.

**FC-to-SC.** In Fig. 7, we first calculate the absolute value of the difference between the prediction and ground truth. Then we binarize them with a defined threshold. The large difference ($>$threshold) is set to dark and small difference is set to light. We can see that the controlled reverse flow (bottom row) shows better prediction result with significantly less difference.

**Task Functional Connectivity Generation** We explain our findings on HCP-YA datasets here. We first record all the predicted functional connectivities from test sets. Then, we also record the ground-truth functional connectivities of the whole dataset. Next, we group them by different tasks and averages within the groups on the predicted (at the top of the Fig. 4) and ground-truth FC (at the bottom of the Fig. 4). Fig. 4 reveals several biologically meaningful patterns that go beyond mere prediction accuracy: (1) **Network Organization Preservation**. We observe block-like structures and prominent diagonal elements that reflect strong intra-hemispheric and inter-regional interactions. (2) **Task-Specific Network Re-organization.** The difference across tasks in the predicted FC matrices reveals that the model is sensitive to the task-induced reconfiguration of functional connectivity, consistent with the literature on dynamic functional networks. For example, in the MOTOR task, we

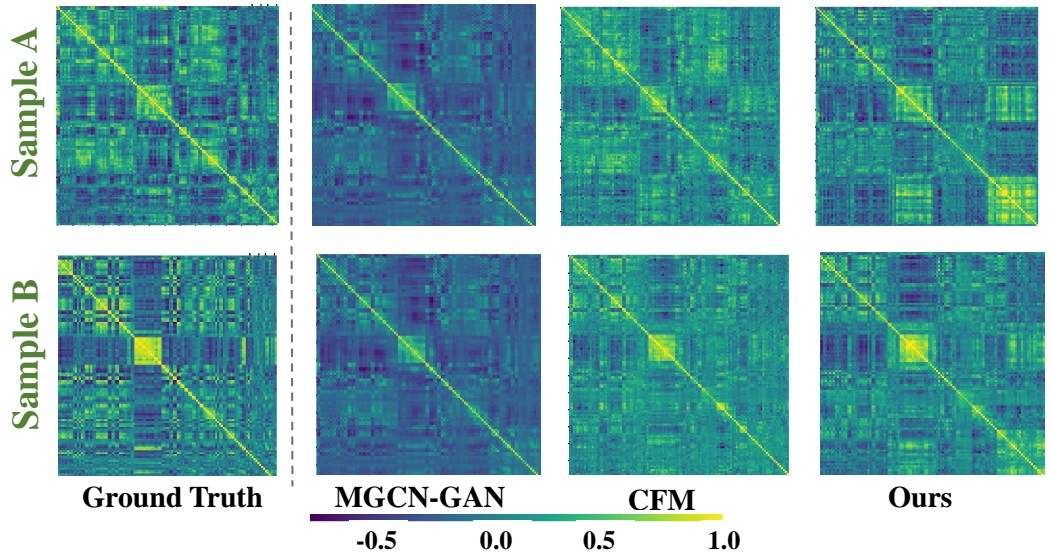

Figure 6: Then predicted results of different methods. Two randomly drawn samples are displayed on two rows.

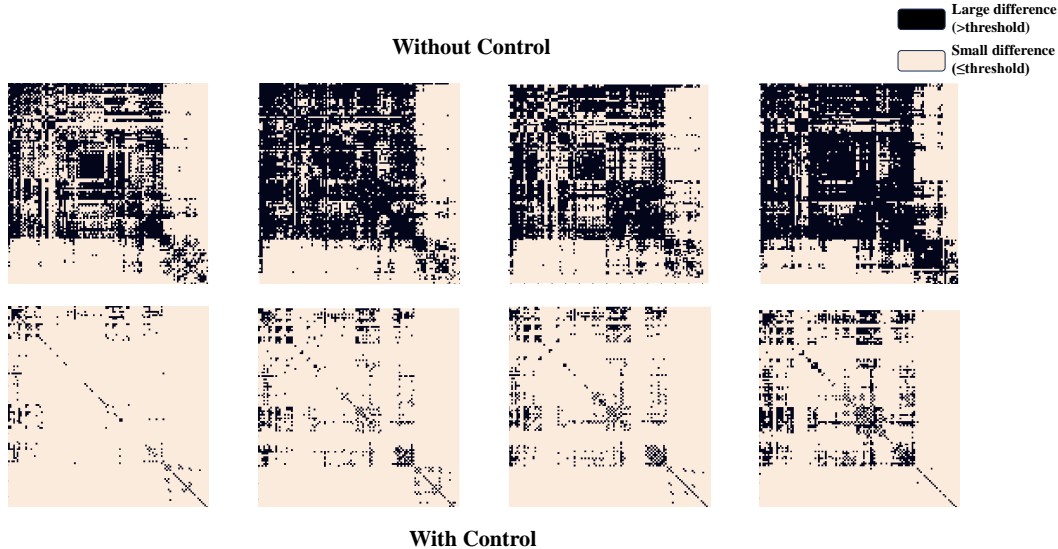

Figure 7: The results of reverse prediction. The large difference (>threshold) is set to dark and small difference is set to light. Each column represents a result predicted from one specific state FC from same subject. The top column is the result without control and the bottom row is the result using consensus control.

can see the strong activation in the top-left part of the matrix (The motor networks are indexed with about 1-20), which aligns with the findings in work by Tzourio-Mazoyer N et.al.[38].

**Evaluation on each Task of HCP-YA.**

*Experimental description.* We conduct task-specific functional connectivity prediction for each of the seven cognitive tasks in the HCP-YA dataset. Using the same experimental setup as described in Sec. 5.2, we train all baseline methods (AM, VPFM, [SF]$^2$M, CFM) and *BrainFlow* to predict task-specific FCs from the same structural connectivity. For each task, we evaluate the prediction quality using three metrics: RMSE for edge-wise accuracy, LCE for local network topology preservation, and GCE for global network structure preservation. All experiments are repeated 5 times with different random seeds, and we report mean $\pm$ standard deviation for each metric-task combination.

*Results.* Table 8 presents the comprehensive task-specific prediction performance across all seven cognitive tasks. *BrainFlow* consistently achieves the best or near-best performance on topological metrics (LCE and GCE) across all seven tasks. Specifically, *BrainFlow* obtains the lowest LCE in all seven tasks, with values ranging from 8.1% (Motor) to 10.1% (Relational), demonstrating superior local network topology preservation. The standard deviations across all metrics remain small (typically <0.3%), indicating stable and reproducible performance across different random initializations. Among the remaining baselines, CFM demonstrates competitive RMSE performance but falls short in preserving network topology compared to *BrainFlow*.

*Discussion.* The task-specific analysis reveals three key insights. First, each method maintains stable performance across all seven cognitive tasks, with small standard deviations indicating consistent behavior regardless of the functional state. Second, the relative ranking of methods remains consistent across tasks: *BrainFlow* consistently achieves the best topological preservation (LCE/GCE), while CFM obtains the lowest RMSE. This stability suggests that model performance is primarily driven by architecture and training objectives rather than task-specific properties. Third, the narrow performance range across tasks (e.g., *BrainFlow*'s LCE varies only from 8.1% to 10.1%) demonstrates that our method has learned a unified SC-FC relationship that generalizes robustly across diverse cognitive states, supporting the hypothesis that a shared structural scaffold underlies multiple functional configurations.

# E Proof

### E.1 Proof of Proposition 4.2

*Proof.* We first introduce *conservation of probability measure* on the Riemannian manifold. Assume a probability density $\rho_0 : \mathcal{M}_0 \to \mathbb{R}+$ defined on a Riemannian manifold $\mathcal{M}_0$ and a transformation $\mathscr{T} : \mathcal{M}_0 \to \mathcal{M}_1$ that maps the Riemannian manifold $\mathcal{M}_0$ to $\mathcal{M}_1$. Then the density $\rho_0$ together with transformation $\mathscr{T}$ induce a probability density $\rho_1 : \mathcal{M}_1 \to \mathbb{R}+$ via the relation:

$$\int_{\mathcal{M}_0} \rho_0(x_0) d\text{vol}_{x_0} = \int_{\mathcal{M}_1} \rho_1(x_1) d\text{vol}_{x_1}$$

According to [24], one can derive:

$$\rho_1(x_1) = \rho_0\left(\mathscr{T}^{-1}(x_1)\right) \left|\det J_{\mathscr{T}^{-1}}(x_1)\right| \tag{23}$$

where $\left|\det J_{\mathscr{T}^{-1}}(x_1)\right|$ means the determinant of the Jacobian of $\mathscr{T}^{-1}$. Here we apply this theorem to Riemannian manifold $(\mathcal{S}_n^+, g)$ and $(\mathcal{L}_n^+, \tilde{g})$ with Cholesky decomposition $\mathscr{L}$, an **isometry** between them. We can directly induce that:

$$\int_{\left(\mathcal{S}_n^+, g\right)} \rho_s(X) d\text{vol}_X = \int_{\left(\mathcal{L}_n^+, \tilde{g}\right)} \rho_l(L) d\text{vol}_L \tag{24}$$

Because $\mathscr{L}$ is an isometry between $(\mathcal{S}_n^+, g)$ and $(\mathcal{L}_n^+, \tilde{g})$, then $\mathscr{L}$ preserves the metric and, consequently, the volume measure. This directly implies:

$$d\text{vol}_X = d\text{vol}_L \ (\text{under the isometry} \mathscr{L})$$

Thus, the Jacobian determinant adjustment $\left|\det J_{\mathscr{L}^{-1}}(L)\right|$ is 1 everywhere (notice that the inverse of an isometry is also an isometry). Therefore,

$$\rho_s(X) = \rho_l(L), \forall \mathscr{L}(X) = L, X \in \mathcal{S}_n^+$$

## E.2 Proof of Theorem 4.3

To enhance the readability of the proof, we omit the input of $V_t(\cdot)$ and $\widetilde{V}_t(\cdot)$. According to **Section 3.2** of [10], the inverse map of the Cholesky decomposition $\mathscr{S} : \mathcal{L}_+ \to \mathcal{S}_n^+$ by $\mathscr{S}(L) = LL^\top$ is an isometry between $(\mathcal{L}_+, \tilde{g})$ and $(\mathcal{S}_n^+, g)$. Specifically,

$$\tilde{g}_L(P, Q) = g_{\mathscr{S}(L)}\left((D_L\mathscr{S})(P), (D_L\mathscr{S})(Q)\right)$$

for all $L \in \mathcal{L}_+$ and $P, Q \in T_L\mathcal{L}_+$. Here the differential $D_L\mathscr{S} : T_L\mathcal{L}_+ \to T_{LL^\top}\mathcal{S}_n^+$ is given by

$$D_L\mathscr{S}(P) = LP^\top + PL^\top$$

Next, again because $\mathscr{S}$ is an isometry and $X_0 = \mathscr{S}(L_0), X_1 = \mathscr{L}(L_1)$, according to **Proposition 5.6.** of [39], the map to the geodesic $\mathscr{S}(L_t)$ between $L_0$ and $L_1$ is the geodesic between $X_0$ and $X_1$, i.e., $\mathscr{S}(L_t) = X_t$.

Now we have

$$\tilde{g}_{L_t}(P, Q) = g_{X_t}\left((D_{L_t}\mathscr{S})(P), (D_{L_t}\mathscr{S})(Q)\right)$$

According to the definition of a norm, we have $\left\|\widetilde{V}_t - \dot{L}_t\right\|_{\tilde{g}_{L_t}}^2 = \tilde{g}_{L_t}(\widetilde{V}_t - \dot{L}_t, \widetilde{V}_t - \dot{L}_t)$. Since $\mathscr{S}$ is an isometry between $(\mathcal{L}_+, \tilde{g})$ and $(\mathcal{S}_n^+, g)$, the metrics satisfy:

$$\tilde{g}_{L_t}(\widetilde{V}_t - \dot{L}_t, \widetilde{V}_t - \dot{L}_t) = g_{X_t}\left((D_{L_t}\mathscr{S})(\widetilde{V}_t - \dot{L}_t), (D_{L_t}\mathscr{S})(\widetilde{V}_t - \dot{L}_t)\right)$$

It can be verified that $D_{L_t}\mathscr{S}$ is a linear map. Then the optimal $\widetilde{V}_t^*$ implies that

$$(D_{L_t}\mathscr{S})(\widetilde{V}_t^* - \dot{L}_t) = (D_{L_t}\mathscr{S})\widetilde{V}_t^* - (D_{L_t}\mathscr{S})\dot{L}_t$$
$$= V_t^* - \dot{X}_t$$

Hence, when we find an optimal $\widetilde{V}_t^*$, we can get $V_t^*$ by $V_t^* = L_t\widetilde{V}_t^{*\top} + \widetilde{V}_t^* L_t^T$ and

$$\tilde{g}_{L_t}(\widetilde{V}_t - \dot{L}_t, \widetilde{V}_t - \dot{L}_t) = g_{X_t}\left(V_t^* - \dot{X}_t, V_t^* - \dot{X}_t\right)$$

Therefore, by the isometry property of $\mathscr{S}$ and Proposition 4.2, the optimization problems in $(\mathcal{L}_n^+, \tilde{g})$ and $(\mathcal{S}_n^+, g)$ are equivalent.

## F  Limitations

Here we discuss the potential limitations of *BrainFlow*. The first point is that though we prove the equivalence of the optimal solution between SPD manifold and Cholesky manifold, the convergence speed is not rigorously discussed. We only evaluate them empirically.

Second, the model shows better performance on larger dataset like UK Biobank compared with the other two datasets, revealing that the method requires large number of data for good performance, which is not easy for neuroimaging data.

Third, we acknowledge the well-known challenge that the relationship between structural and functional connectivity is not a simple one-to-one mapping at the edge level. Complex biological mechanisms, such as polysynaptic pathways and neuromodulatory effects, mean that strong functional links can exist between regions with no direct anatomical connection. Furthermore, the inherent limitations of diffusion MRI and tractography algorithms can lead to both false positives and false negatives in SC estimation. Our model does not explicitly simulate these underlying biological mechanisms. Instead, BrainFlow addresses this challenge by learning a mapping between the entire distribution of structural connectomes and the distribution of functional connectomes. By leveraging paired SC-FC data from the same individuals to construct a joint distribution, the model captures the high-level statistical regularities that emerge from these complex, indirect interactions. Therefore, BrainFlow's strength lies in its ability to learn and represent the statistical SC-FC coupling within the available imaging data, thereby contributing to our understanding of this well-known mismatch.

# G Impact Statement

This paper advances the field of Machine Learning by introducing *BrainFlow*, a generative model that captures the coupling between structural and functional brain connectivity. By leveraging flow-based optimization on the SPD manifold, our approach enhances the interpretability and robustness of learning structured and dynamic data. While our work has potential implications for neuroscience and AI, there are no specific societal consequences that must be highlighted here.

