# OpenReview forum: "BrainFlow: A Holistic Pathway of Dynamic Neural System on Manifold"
_NeurIPS.cc/2025/Conference — NeurIPS 2025 poster_

### Official Review · Reviewer_5mrG · 2025-07-02

**Clarity:** 4
**Significance:** 3
**Originality:** 3
**Rating:** 5
**Confidence:** 4

**Summary:**

BrainFlow proposes a reversible, continuous-time generative model on the SPD manifold, learning an ODE-driven flow to transform structural connectivity into state-specific functional connectivity and back within a single framework. It leverages an isometric embedding into the Cholesky manifold for computational efficiency and adds a Dirichlet-energy consensus term to ensure multi-state SC–FC consistency.

**Questions:**

Adding to the concerns raised on the weaknesses section, I have the following questions

* Would jointly learning σ (consensus weight) improve performance?

* How sensitive are results to the choice of consensus weight σ? Could the authors provide an ablation study varying σ and its impact on reconstruction error?

* Could the authors comment on if/how model interpretability has informed any neuroscientific insights?

**Ethical Concerns:**

["NO or VERY MINOR ethics concerns only"]

**Final Justification:**

The authors have addressed most of the concerns,

**Limitations:**

yes

**Quality:**

3

**Strengths And Weaknesses:**

Strengths

* Optimizing on the Cholesky manifold is a clever way to guarantee that all geometric and probabilistic properties are preserved with significantly lower computational overhead.

* The consensus control nicely embeds multi‐state biological consistency directly into the dynamics.

* The paper has a nice flow with solid theoretical grounding.

Weaknesses

* To fit structural connectivity into the SPD manifold, a “small” self-loop is added; however, the authors do not clarify how the choice of self-loop magnitude affects downstream flows or biological interpretability.

* The Dirichlet-energy consensus term is tacked on during reverse ODE integration (eq. (14)) rather than baked into training. It would be interesting for the authors to verify whether this creates a mismatch between the learned vector field and its application at test time.

* The comparisons focus on GAN- and flow-based methods. More recent diffusion/score-based generative models [1,2]  might offer competitive performance and would strengthen the empirical evaluation if included.

[1] Li, Yunchen, et al. "Spd-ddpm: Denoising diffusion probabilistic models in the symmetric positive definite space." Proceedings of the AAAI Conference on Artificial Intelligence. Vol. 38. No. 12. 2024.
[2] De Bortoli, Valentin, et al. "Riemannian score-based generative modelling." Advances in neural information processing systems 35 (2022): 2406-2422.

---

> ### Author Rebuttal · Authors · 2025-07-29
>
> # W1
> + We choose the value of \epsilon we used is fixed to be the largest value in the training set such that we can recover the prediction of the structural connectivity to the __original range__. Hence this operation will not affect the biological interpretability and is just for training stability.
>
> # W2, Q1, Q2
> + __This “mismatch” wais intentional and aligns with our design objectives.__ As noted in the main text, vanilla reverse flow introduces integration errors during prediction. Consensus control specifically addresses this by using complementary information across task FCs to reduce such errors at test time. Therefore, the apparent mismatch between training (simulation-free) and inference (with consensus control) is intentional—it allows us to correct for the limitations of the learned vector field during application.
> + We also performed sensitivity analysis on the consensus control weights $\sigma$ to validate this strategy. We can see that when $sigma$ is small (~0.2), the model has significant performance gain compared to vanilla reverse flow. However, as $\sigma$ increases, the consensus control comes to dominate the reverse flow, causing the trajectories to become biased.
> + We then argue that $\sigma$ may not be suitable to optimize during training.
>    + First, the reversible nature of our flow means that Dirichlet energy regularization applied to the reverse flow also constrains the forward flow, thereby reducing the diversity of predicted task FCs.
>    + Second, there is a mismatch between training and inference objectives. While consensus control leverages complementary information across different task FCs to reduce simulation error during inference, BrainFlow training is simulation-free and does not account for this simulation error. Therefore, optimizing σ during training is unsuitable for the intended inference procedure.
>
>
> +  __HCP-A__
> | $\sigma$ | 0.0| 0.2| 0.4 | 0.6 | 0.8 |
> | :--- | :---: | :---: | :---: | :---: | :---: |
> | MSE | 0.65±0.00 | 0.13±0.00 | 0.44±0.01 | 1.52±0.03 | 2.37±0.01 |
> | LCE(%) | 0.74±0.00 | 0.71±0.00 | 0.95±0.00 | 3.17±0.03 | 20.28±0.05 |
> | GCE(%) | 0.73±0.00 | 0.69±0.00 | 0.93±0.01 | 3.18±0.02 | 20.50±0.03 |
>
> + __HCP-YA__
> | $\sigma $ | 0.0| 0.2| 0.4 | 0.6 | 0.8 |
> | :--- | :---: | :---: | :---: | :---: | :---: |
> | MSE | 0.42±0.00 | 0.14±0.00 | 0.52±0.00 | 1.32±0.02 | 2.12±0.01 |
> | LCE(%) | 0.62±0.00 | 0.55±0.00 | 0.93±0.00 | 2.92±0.01 | 15.74±0.03 |
> | GCE(%) | 0.59±0.00 | 0.59±0.00 | 0.88±0.00 | 3.02±0.02 | 16.40±0.02 |
>
> + __UKB__
> | $\sigma$ | 0.0| 0.2| 0.4 | 0.6 | 0.8 |
> | :--- | :---: | :---: | :---: | :---: | :---: |
> | MSE | 0.09±0.00 | 0.07±0.00 | 0.69±0.00 | 1.89±0.002 | 2.52±0.01 |
> | LCE | 0.52±0.00 | 0.39±0.00 | 1.33±0.01 | 10.93±0.02 | 27.88±0.02 |
> | GCE | 0.53±0.00 | 0.39±0.00 | 1.30±0.01 | 11.16±0.01 | 28.16±0.02 |
>
> # W3
> + We appreciate the reviewer's suggestion to include diffusion/score-based methods. However, there are fundamental differences in problem formulation that make direct comparison challenging.
> + Diffusion models are designed for generation from noise distributions to target data distributions, whereas our task requires deterministic transport between two specific, biologically-coupled distributions. We need to map from a given structural connectivity (SC) to its corresponding functional connectivity (FC) from the same individual, preserving subject-specific relationships rather than generating samples from scratch. Thus, diffusion models are not included in the comparison.
> + Besides, we include Schrodinger Bridge based method, which is a variant of diffusion model that can transfer between distributions into our experiments. We believe our current comparison provides an appropriate baseline given the transport nature of our problem.
>
> # Q3
> + Our model's design is directly informed by neuroscientific principles to ensure its interpretability. For instance, the consensus control feature is based on the biological reality that different cognitive functions rely on shared neural circuits anchored by the stable underlying structural connectivity (SC). This ensures our model's predictions are consistent and biologically plausible.
> + The model also captures how functional networks dynamically reconfigure for different cognitive demands, successfully generating meaningful, task-specific connectivity patterns that align with established neuroscience literature.

---

> ### Comment · Reviewer_5mrG · 2025-08-05
>
> I thank the authors for their thorough responses. Their explanations have mostly addressed my concerns and as a result I am increasing my score.

---

> > ### Author Response · Authors · 2025-08-05
> >
> > We sincerely appreciate that.

---

### Official Review · Reviewer_67by · 2025-07-02

**Clarity:** 4
**Significance:** 3
**Originality:** 4
**Rating:** 5
**Confidence:** 3

**Summary:**

This paper introduces BrainFlow, a generative model for learning a reversible mapping between structural (SC) and functional connectivity (FC) in the brain. It uses flow matching on the Riemannian manifold of symmetric positive-definite (SPD) matrices and optimizes on the Cholesky manifold for efficiency. A consensus control mechanism improves reverse flows by aligning FC states from the same subject. Experiments are conducted on synthetic data and three large neuroimaging datasets: UK Biobank, HCP-Aging, and HCP-Young Adult.

**Questions:**

The model uses subject-paired SC-FC training pairs, but are those pairs treated independently during training?

Is the consensus only used for reverse inference, or could it play a regularizing role in training as well?

The ethics checklist claims the study does not involve human subjects. This is incorrect, as secondary analysis of human neuroimaging data qualifies as human subjects research. Depending on local policies, secondary analyses may or may not be exempt from ethics approval, but this is still regulated human research. UK biobank access requirements in particular are quite strict and accessing these data now requires mandatory ethics training. Please clarify this aspect of the submission.

**Ethical Concerns:**

["NO or VERY MINOR ethics concerns only"]

**Final Justification:**

After reading the rebuttal and the other reviews, I am comforted in my positive evaluation of this work.

My initial assessment identified the technical description of neuroimaging experiments as a weak point, which the authors fully addressed in their rebuttal. This led to increase my score on clarity to 4.

I was also unsure about the choice of baselines. This was also clarified by the rebuttal, and I now think there is a strong mixture of reference models and tasks for the models proposed in the paper.

I would have liked to see more rigorous experiments on the data efficiency of the methods, but the authors have expanded the range of applications to include other types of datasets with varied dimensions, so I do not see this point as a real weakness.

Overall, I believe this work to be a substantial contribution to the field, with strong methodological and experimental original developments, and I recommend acceptance.

**Limitations:**

## Speculative Claims About Scaling
The performance gain on UK Biobank is attributed to dataset size without a scaling experiment. The opportunity to explore scaling across task complexity in HCP datasets is missed.

**Paper Formatting Concerns:**

## Readability issues
* Section 2.2 is dense and hard to follow.
* Some notation (e.g., (_{1/2})) is poorly introduced.
* Minor formatting issues (e.g., missing spaces) recur.
* The Related Work section is oddly placed at the end, and the models included in the benchmark are not all introduced.

**Quality:**

4

**Strengths And Weaknesses:**

# Strengths
 * BrainFlow addresses the problem of SC-FC coupling across tasks using a geometrically principled and computationally efficient approach, that truly models inter-subject and inter-task variance in the domains of structure and function simultaneously.
* The model is bidirectional and incorporates a creative consensus control mechanism.
* Results are evaluated on large-scale datasets with a variety of metrics and relevant baselines.
* Figures are informative and suggest strong qualitative preservation of structure.
* The paper seems honest in its reporting.

# Weaknesses
## Misleading brain connectivity terminology
Terms like "dendritic coupling mechanism" are misleading: MRI does not provide dendritic-level information. SC is described as "number of nerve fibers," which oversimplifies tractography. The terminology should be more rigorous, and authors should outline what dMRI derivatives they are using (tractography counts?).


## Missing Details on Connectivity and Dataset Composition
FC metric construction (e.g., correlation type, GSR, regularization) is unspecified. SC pipeline details (e.g., tractography method, filtering) are missing. The number of usable SC-FC pairs in HCP-YA is unclear; only ~300 subjects have DWI.


## Evaluation Framing Issues
Table 2 lacks the classifier's performance on real FCs, making interpretation difficult (although this is indicated in the text for UK biobank). It’s unclear if other models in Table 2 were conditioned on task; MGCN-GAN is missing.

# Conclusion
Overall assessment: BrainFlow combines an elegant theoretical framework with competent empirical work and a welcome level of transparency. Despite weaknesses in domain framing, missing methodological detail, and some presentation issues, the paper advances the modeling of SC-FC relationships in a meaningful way. I recommend acceptance, provided revisions improve clarity and align biological claims with the data.

---

> ### Author Rebuttal · Authors · 2025-07-29
>
> # W1
> + We apologize for the confusion. We will rephrase the technical terms.
> + The measurement we use to construct structural connectivity is the fiber count.
>
> # W2
> + For functional connectivity preprocessing and construction, we use xcp-d, a post-processing pipeline for fMRI data [1].
> + As for the structural connectivity, we utilize QSIPrep [2] for processing and reconstruction.
> + XCP-D computes functional connectivity matrices using Pearson correlation coefficients between parcellated time series. The pipeline employs the 36P nuisance regression strategy, which includes six motion parameters along with their derivatives and quadratic terms, plus white matter, cerebrospinal fluid, and global signal regressors. The correlation calculations themselves do not include any explicit regularization beyond the standard nuisance regression approach.
> + QSIPrep's default reconstruction workflows use iFOD2 (2nd-order Integration over Fiber Orientation Distributions) as the tractography algorithm, and SIFT2 (Spherical-deconvolution Informed Filtering of Tractograms 2) is enabled to provide biologically meaningful connection weights.
> + We apologize for the confusion on  image demographic information. We clarify that the usable SC-FC pair is 3,592, which is mainly constrained by the number of DWI. The remaining fMRI that don't have corresponding DWI are deprecated.
> + We will include all information into the supplement material of the final version.
> # W3
> + We would like to gently note that Table 2 has included metrics for the classifier’s performance on real FCs from other datasets, alongside the details mentioned in the text for UK Biobank.
> + To avoid any confusion, we will revise the table caption in the revised manuscript to explicitly highlight this inclusion, ensuring the information is more immediately apparent.
> + And other models in Table 2 were conditioned on task with flexible class guidance. However, as the original MGCN-GAN just focused on resting FC instead of task-specific prediction. We didn't include it in the task recognition experiment.
> # Q1
> + Each pair is created according to subject constraint. During training, each pair is sample from the joint distribution, i.e. $(X_0, X_1) \sim q(X_0,X_1)$ instead of $(X_0,X_1)\sim p(X_0)p(X_1)$. Each sampled pair are considered independent.
> # Q2
> + We argue that consensus control may not be suitable to regularize the training.
>   + First, the reversible nature of our flow means that Dirichlet energy regularization applied to the reverse flow also constrains the forward flow, thereby reducing the diversity of predicted task FCs.
>   +  Second, there is a mismatch between training and inference objectives. While consensus control leverages complementary information across different task FCs to __reduce simulation error__ during inference, BrainFlow training is __simulation-free__ and does not account for this simulation error. Therefore, using the consensus control to regularize the training may not be a good choice.
> # Q3
> + We will change to “involve human subject”. All authors have the HIPPA and CITI training certificates and qualified for processing (de-identified) neuroimaging data.
> # Limitations
> We admit that the "larger dataset" claim for UK Biobank performance is speculative. As the performance gain may also be attributed to the simpler task states (only two different states). Different scanning protocols, populations, and task types across datasets may also be the factors that influence the performance. We will clarify these claims more clearly.
> # Readability issues
> + We will add more intuitive explanation with less mathematical language in Section 2.2
> + We will add a subsection for notation introduction to improve the readability.
> + We will correct the formatting issuses.
> + We will place the related work before experiments and then include all the baseline methods in the Related Work section.
> ---
> [1] Mehta, K., Salo, T., Madison, T., Adebimpe, A., Bassett, D. S., Bertolero, M., Cieslak, M., Covitz, S., Houghton, A., Keller, A. S., Luo, A., Miranda-Dominguez, O., Nelson, S. M., Shafiei, G., Shanmugan, S., Shinohara, R. T., Sydnor, V. J., Feczko, E., Fair, D. A., & Satterthwaite, T. D. (2023). XCP-D: A Robust Pipeline for the post-processing of fMRI data. bioRxiv : the preprint server for biology, 2023.11.20.567926. https://doi.org/10.1101/2023.11.20.567926
>
> [2] Cieslak, M., Cook, P.A., He, X. et al. QSIPrep: an integrative platform for preprocessing and reconstructing diffusion MRI data. Nat Methods 18, 775–778 (2021). https://doi.org/10.1038/s41592-021-01185-5

---

> > ### Comment · Reviewer_67by · 2025-08-04
> >
> > I thank the authors for their detailed and constructive rebuttal.
> >
> > I appreciate the clarifications on the processing pipelines and connectivity metrics used, as well as the details on usable samples. These responses fully address W1 and W2.
> >
> > Regarding W3, apologies for missing some of the information, your efforts to further improve presentation are appreciated. The rationale for excluding MGCN-GAN is reasonable and could be briefly noted in the text.
> >
> > Thank you also for clarifying the ethical compliance. It’s good to know that all authors have the proper training and credentials, and that this will be reflected in the final submission.
> >
> > Finally, with respect to scaling, softening the conclusions is appropriate. Still, a direct experiment on the impact of dataset size or task complexity would have strengthened the paper.
> >
> > Overall, the rebuttal addresses most of my original concerns and reinforces my position in favour of acceptance.

---

> > > ### Author Response · Authors · 2025-08-05
> > >
> > > We sincerely thank the reviewer for the thoughtful and encouraging feedback. We’re glad that our clarifications addressed W1–W3 and appreciate the suggestion to briefly mention the MGCN-GAN exclusion rationale in the text — we will incorporate that in the final version. We also acknowledge the point regarding dataset scaling and will soften the conclusions accordingly. Thank you again for your support and constructive input.

---

### Official Review · Reviewer_mEsD · 2025-07-03

**Clarity:** 3
**Significance:** 2
**Originality:** 3
**Rating:** 4
**Confidence:** 5

**Summary:**

This paper introduces BrainFlow, a reversible flow-based generative model that bridges structural connectivity (SC) and functional connectivity (FC) distributions via conditional flow matching on the symmetric positive definite (SPD) manifold. The method leverages the Cholesky manifold as a computationally efficient surrogate for the SPD manifold, ensuring geometric fidelity and stability. A key novelty is the consensus control mechanism, which harmonizes the reverse flow across multiple FC states to produce structurally coherent outputs. The approach is validated on synthetic data and large-scale neuroimaging datasets (HCP-YA, HCP-A, UK Biobank), demonstrating strong performance on both SC-to-FC and FC-to-SC mappings, with a particular advantage in preserving topological properties of brain networks.

**Questions:**

Questions
Visualization of Output: Can the authors provide explicit examples of SC–to–FC mappings across multiple tasks? It would be helpful to see how one SC gives rise to several FCs and whether those match known task-specific patterns.

Biological Gaps in SC–FC Modeling: How does BrainFlow handle functional relationships not directly observable in SC data (e.g., polysynaptic or modulatory pathways)? Has the model been evaluated on such cases where the SC–FC coupling is weak?

Consensus Control Sensitivity: How sensitive is performance to the consensus control parameter σ? Could this be optimized during training rather than set manually?

Task State Generalization: Does the model generalize to unseen cognitive states or does it require retraining for each state? Can it interpolate between known task-specific FCs?

Dynamic FC Extension: Can BrainFlow be extended to model time-varying FC trajectories? This would seem a natural next step given the flow-based foundation.

Score Increase Criteria: Inclusion of clear visualizations of estimated SC–FC mappings, and a discussion or empirical treatment of cases where SC underrepresents functional pathways, would significantly improve interpretability and trust in biological validity.

**Ethical Concerns:**

["NO or VERY MINOR ethics concerns only"]

**Final Justification:**

Thanks for the detailed response. Most of my concerns are addressed. Given that the original paper lacks visualizations on connectivities (I understand no figures can be provided during rebuttal, however, I expect to see these as they can be important to justify the model's plausibility), I will maintain my score.

**Limitations:**

No major omissions. However, the paper could better address the potential disconnect between SC and FC due to neuroimaging limitations, and clarify when the model’s assumptions may not hold.

**Paper Formatting Concerns:**

no concerns

**Quality:**

4

**Strengths And Weaknesses:**

Strengths:

Originality: Introduces a novel bidirectional model between SC and FC using flow matching on the Cholesky manifold, a less-explored but mathematically rigorous space. The integration of consensus control is a biologically meaningful enhancement.

Technical Quality: The paper rigorously defines the mathematical framework, includes theoretical guarantees (e.g., isometry between SPD and Cholesky manifolds), and provides closed-form velocity expressions and optimization procedures.

Significance: Addresses a central problem in cognitive neuroscience: modeling the SC–FC relationship. The reversible formulation allows both predictive modeling and insight into underlying mechanisms.

Empirical Validation: Experiments on real datasets with various evaluation metrics (RMSE, clustering error, classification accuracy) convincingly support the model’s claims. The method achieves the best performance in preserving topological properties.

Weaknesses:

Lack of Visualization of Learned Connectivities: Despite the focus on modeling mappings between SC and FC, the paper provides limited visualization of the actual predicted connectomes. Figures illustrating specific examples of how a single SC gives rise to distinct FC patterns across cognitive states would help demonstrate the biological plausibility and interpretability of the model's outputs.

SC–FC Mismatch in Biology: The assumption that FC can be reconstructed from SC (and vice versa) may be overly strong. In practice, many FC relationships—especially those reflecting long-range integration or indirect modulation—are not directly observable in structural scans due to tractography limitations. For instance, interhemispheric and long-range connections are frequently underestimated. The paper could benefit from a more explicit discussion of these biological mismatches and whether the model compensates for them.

Clarity for Neuroscience Audience: While the model is technically sound, the biological interpretation of flows and consensus control could be elaborated further for neuroscientists less familiar with manifold-based generative modeling.

Baseline Limitations: Some recent deep learning baselines (e.g., graph neural networks tuned for dynamic FC prediction) are not compared. The study primarily benchmarks against flow-based or conventional methods.

Task Classification Evaluation: The use of pre-trained classifiers for evaluating generated FC could benefit from cross-validation or additional controls to ensure robustness across tasks and datasets.

Generalization to Dynamic FC: Although briefly mentioned as future work, the current approach is limited to static FCs. Its extension to time-varying FC would be important for broader applicability.

---

> ### Author Rebuttal · Authors · 2025-07-29
>
> # W1, Q1
> + We agree with the reviewer that the visualization from the same structural connectivity to different task functional connectivity is missing. We will update the corresponding figure in the final manuscript as external links are not allowed.
> # W2, Q2
> + We appreciate this very insightful comment.
>   + At the edge level, we completely agree that the relationship between structural and functional connectivity is far from a simple one-to-one mapping, due to the biological mechanisms mentioned by this reviewer. To address this challenge, we consider each SC and FC matrix as a data instance on the manifold. In this context, the collection of SC and FC matrices form the distribution on the manifold. Our method is trained to learns a mapping from the entire structural connectome distribution to the entire functional connectome distribution.
>   + Recall that during training, it leverages __paired SC-FC__ data from the same subject to construct a joint distribution, enabling the flow to capture statistical properties of __both structural and functional connectivity__ including those arising from indirect pathways that manifest in FC but are not explicitly encoded in SC.
>   + To better address this important point in our manuscript, we will add a paragraph to the Limitations section (Appendix F). This new text will explicitly acknowledge the challenges of the SC-FC mismatch and tractography limitations. We will clarify that _BrainFlow_'s strength lies in its ability to learn the statistical regularities within the available imaging data and thereby contributing to understanding this well-known mismatch.
>
> # W3
> + We appreciate this constructive comment. To improve clarity, we will integrate the concise interpretation of BrainFlow. For example, a "flow" in our model is not a physical process but a learned transformation pathway like interpolation between SC and FC. Consensus control enforces the biological principle of a shared structural scaffold. The brain's stable anatomy (SC) supports many dynamic functional states (FCs). We will integrate these explanations into the __Introduction__ and __Methods__ sections to improve the manuscript's accessibility.
>
> # W4
> + We agree with this reviewer’s opinion. We therefore add a robust graph neural network baseline, GNN$^+$ [1], which showed excellent performance on most graph-level tasks with simple architecture. We set the objective as the MSE loss. Other details will be displayed on the final version. The result shows that  GNN$^+$ performs consistently across three datasets, with superior RMSE and relatively high GCE/LCE. However, another observation from the visualization (not allowed during rebuttal) is that most of the predictions of GNN$^+$ (i.e. the non-flow method) show very similar patterns. We argue that this is because the MSE loss drives predictions toward the center of the training distribution (similar to regression to the mean), minimizing overall training loss but sacrificing individual-specific patterns. This results in good aggregate metrics but poor preservation of subject-level variability, as evidenced in the visualization.
> + ||HCP-A RMSE | HCP-A GCE | HCP-A LCE | HCP-YA RMSE  | HCP-YA GCE | HCP-YA LCE   | UKB RMSE    | UKB GCE     | UKB LCE     |
> | :--- | :----------- | :----------- | :----------- | :----------- | :----------- | :----------- | :----------- | :----------- | :----------- |
> | GNN$^+$ | 0.211±0.000  | 13.4±0.0  | 10.4±0.2  | 0.242±0.001  | 13.7±0.1  | 10.3±0.0 | 0.216±0.001  | 13.8±0.0  | 9.9±0.0 |
> # W5
> + We apologize for the confusion. In our experiment, we have applied a very strict validation plan. Specifically, the pre-trained classifier was first trained on 5-fold cross-validation to select hyperparameters of the model. We then use the whole training set to train this classifier. Finally we classify the generated samples.
> # W6, Q5
> + We completely agree with this reviewer. In this work, we put the spotlight on the method development of solving the foundational "one-to-many" mapping problem, i.e., bridging a single static structural connectome (SC) to multiple, distinct static functional connectomes (FCs) associated with different cognitive states. However, the "flow" concept at the core of our model is highly general and well-suited for modeling various types of neural dynamics.
> + In macro-scale, BrainFlow framework could be used to model the slow evolution of brain connectivity over months or years (lifespan).
> + In micro-scale, the framework can be adapted to model rapid, within-scan fluctuations in functional connectivity. In this context, the model would learn a flow that maps an FC matrix from one time window to the next.
> # Q3
> + We conduct an experiment to test the sensitivity of the consensus control parameter σ. We can see that when $\sigma$ is small (~0.2), the model has significant performance gain compared to vanilla reverse flow. However, as $\sigma$ increases, the consensus control comes to dominate the reverse flow, causing the trajectories to become biased.
> + We then argue that $\sigma$ may not be suitable to optimize during training.
>   + First, the reversible nature of our flow means that __Dirichlet energy regularization__ applied to the reverse flow also constrains the forward flow, thereby __reducing the diversity__ of predicted task FCs.
>   + Second, there is a mismatch between training and inference objectives. While consensus control leverages complementary information across different task FCs to __reduce simulation error__ during inference, BrainFlow __training is simulation-free__ and does not account for this simulation error. Therefore, optimizing $\sigma$ during training is unsuitable for the intended inference procedure.
> +  __HCP-A__
> | $\sigma$ | 0.0| 0.2| 0.4 | 0.6 | 0.8 |
> | :--- | :---: | :---: | :---: | :---: | :---: |
> | MSE | 0.65±0.00 | 0.13±0.00 | 0.44±0.01 | 1.52±0.03 | 2.37±0.01 |
> | LCE(%) | 0.74±0.00 | 0.71±0.00 | 0.95±0.00 | 3.17±0.03 | 20.28±0.05 |
> | GCE(%) | 0.73±0.00 | 0.69±0.00 | 0.93±0.01 | 3.18±0.02 | 20.50±0.03 |
>
> + __HCP-YA__
> | $\sigma$ | 0.0| 0.2| 0.4 | 0.6 | 0.8 |
> | :--- | :---: | :---: | :---: | :---: | :---: |
> | MSE | 0.42±0.00 | 0.14±0.00 | 0.52±0.00 | 1.32±0.02 | 2.12±0.01 |
> | LCE(%) | 0.62±0.00 | 0.55±0.00 | 0.93±0.00 | 2.92±0.01 | 15.74±0.03 |
> | GCE(%) | 0.59±0.00 | 0.59±0.00 | 0.88±0.00 | 3.02±0.02 | 16.40±0.02 |
>
> + __UKB__
> | $\sigma$ | 0.0| 0.2| 0.4 | 0.6 | 0.8 |
> | :--- | :---: | :---: | :---: | :---: | :---: |
> | MSE | 0.09±0.00 | 0.07±0.00 | 0.69±0.00 | 1.89±0.002 | 2.52±0.01 |
> | LCE | 0.52±0.00 | 0.39±0.00 | 1.33±0.01 | 10.93±0.02 | 27.88±0.02 |
> | GCE | 0.53±0.00 | 0.39±0.00 | 1.30±0.01 | 11.16±0.01 | 28.16±0.02 |
>
> # Q4
> + BrainFlow can not directly generalize to unseen cognitive states. However, based on the trained model, one can easily extend to a new task state by inserting a new task state embedding and optimizing with new data.
> + Similarly, interpolation between known task-specific FCs can not be directly achieved. We can do that by training a new model.
> + It would be interesting to embed the connectivity (both SC and FC) into a unified latent space to achieve flexible transformation.
>
> ## Limitations
> + We clarify that the model may fail when functional connectomes (FCs) from the same subject and task exhibit significantly distinct patterns, rendering it difficult for the model to capture the underlying dependencies of SC-FC pairs.
> ---
> [1] Luo Y, Shi L, Wu X M. Can Classic GNNs Be Strong Baselines for Graph-level Tasks? Simple Architectures Meet Excellence[C] //Forty-second International Conference on Machine Learning

---

> > ### Comment · Reviewer_mEsD · 2025-08-07
> > **Response to Rebuttal by Authors**
> >
> > Thanks for the detailed response. Most of my concerns are addressed. Given that the original paper lacks visualizations on connectivities (I understand no figures can be provided during rebuttal, however, I expect to see these as they can be important to justify the model's plausibility), I will maintain my score.

---

### Official Review · Reviewer_nEq7 · 2025-07-03

**Clarity:** 2
**Significance:** 2
**Originality:** 2
**Rating:** 4
**Confidence:** 4

**Summary:**

The authors propose a novel way to connect subject-specific functional connectivity (FC) and structural connectivity (SC) in MRI data. The authors model SC and FCs as elements of the SPD manifold, overcoming the common Euclidean assumption. The authors provide a computationally efficient method to learn the flow from SC to FCs by proving equivalence of SPD and Cholesky manifolds. Using Dirichlet energy to constrain the backwards flow from multiple FCs to SC, the authors are effectively able to learn the correct subject-specific SC.

**Questions:**

Line 520: What is the \epsilon value used? Was it subject-specific?

Figure 4 right panel: shows a very interesting way the model has chosen to transport from the green circle to the blue – i.e., move to the center first and then to the blue circle. Could the authors comment on this or provide a reasoning why this could be happening?

A suggestion on Section D.3 – HCP-YA: Conditioning on the subtask (instead of the task) could be useful in informing finer details in the SC maps

**Ethical Concerns:**

["NO or VERY MINOR ethics concerns only"]

**Final Justification:**

I have read the authors' rebuttal responses and discussed these responses, am happy to upgrade my score.

**Quality:**

2

**Strengths And Weaknesses:**

Strengths:

1. The proposed method – BrainFlow – is well-motivated and appears to be mathematically well-grounded

2. Proving the equivalence between optimizing on the Cholesky manifold and SPD manifold opens up avenues for other researchers to use the computationally efficient version

3. BrainFlow performs well on graph-based metrics (quantified with LCE and GCE) in addition to RMSE, suggesting that it is able to capture intended patterns

Weaknesses:

1. The evidence for computational efficiency is sparingly mentioned

2. Eq. 14: Is the \sigma the same as the one used in a_{ij}? If so, why?

3. Section D.3 – HCP-YA: Why was resting-state excluded (unlike in HCP-A)?

4. Section D.3 – HCP-YA: It is well-known that FC estimation is affected by run time. Since different tasks have different run times, did the authors perform any correction? If not, do the run times affect the quality of predicted FCs? Prediction quality (RMSE, LCE, GCE) per task would be a valuable add to the manuscript.

5. Line 46: “evolving” may not be the right term since temporal changes in FCs are not modelled

6. Figure 2: Cholsesky -> Cholesky

7. Line 83: Mention S_n^+ before use

8. Line 96: Missing element notation

9. Equation 4: Mention q(z) before use

10. Figure 4: Using different colors for circle and dots (orange) will improve clarity

11. Line 221: typical convention is to use \sigma^2 in Gaussian kernels

12. Line 225: gradient is w.r.t L_1^{y_i}

13. Title should read "Manifold" (it now says "Mainfold")

---

> ### Author Rebuttal · Authors · 2025-07-29
>
> # W1:
> + We will give a theoretical analysis on computational efficiency, as follows, in the final version.
> + First, the flow-based methods can be summarized with the following steps:
>   + Sampling from the source and target distributions.
>   + Calculate the interpolated position and velocity between the source and target samples
>   + Regress the predicted velocity to ground truth velocity.
>   + During inference, integrate the time-dependent velocity.
> + We show the additional computation of Riemannian conditional flow matching on SPD manifold (**RCFM-SPD**) and ___BrainFlow___ on these steps compared to baseline vanilla Flow Matching.
>   + Before sampling stage, additional preprocessing (Cholesky decomposition) is required for BrainFlow, whose time complexity is $\mathcal{O}(Nd^3), N$ is the size of training set.
>   + During interpolation stage, the $\exp(\cdot),\log(\cdot)$ of **RCFM-SPD** both requires $\mathcal{O}(Bd^3)$, where $B$ is the batch size, while these operations on Cholesky manifold only requires $\mathcal{O}(d)$
>   + During loss calculation inner product $<\cdot,\cdot>$ on SPD manifold requires $\mathcal{O}(Bd^3)$ while Cholesky manifold has the same complexity as vanilla Flow Matching ($L_2$ norms)
>   + During inference, **RCFM-SPD** requires eigenvalue clipping to keep the element SPD, requiring $\mathcal{O}(Bd^3)$ each integration step. Hence, it requires additional $\mathcal{O}(LBd^3)$ where $L$ is total step.
> | Stage | RCFM-SPD | BrainFlow | Flow Matching |
> |--|--|--|--|
> | **Sampling** | Standard | + $\mathcal{O}(Nd³)$ Cholesky preprocessing | Baseline |
> | **Interpolation** | +$\mathcal{O}(Bd³)$ $\exp(·), \log(·)$ (**SPD**)| $+\mathcal{O}(Bd)$ $\exp(·), \log(·)$ (**Cholesky**)  | Baseline |
> | **Loss** | +$\mathcal{O}(Bd³)$ (**SPD inner product**) | Standard (**Cholesky inner product**) | Baseline |
> | **Inference** | +$\mathcal{O}(LBd³) $eigenvalue clipping | Standard | Baseline |
> - N: training set size, B: batch size, d: matrix dimension, L: integration steps
> - "+": additional computational cost compared to baseline
> - The computational cost of __Cholesky preprocessing__ is incurred __only once__, whereas the costs associated with __interpolation, loss calculation, and inference__ need to be __repeated at each step__.
> # W2:
> +  $\sigma$ in Eq. 14 and that in $a_{ij}$ are different., We will correct it by changing the coefficient in eq.14. to \beta.
> # W3:
> + Although HCP-A includes both resting-state and task FC data, the dataset is __predominantly composed of resting-state FC__ (four runs per subject) compared to task FC (one run per subject). In the experiment in Section 4.2, we specifically focus on task-evoked connectivity patterns, which provide more controlled experimental conditions. Therefore, we chose to utilize the HCP-YA dataset, which offers a more comprehensive collection of task-based FC data, to validate our condition-related experimental hypotheses.
> # W4, Q3:
> We process the task fMRI by fmriprep and xcp-d, which do not include run time correction. Hence, we follow the reviewer’s idea to do experiments to evaluate the performance per task.  We can see that each method maintains relatively stable performance across all tasks despite varying run times. What’s more, the relative ranking of methods remains consistent regardless of task, suggesting that run time differences do not substantially alter the fundamental performance characteristics of each approach.
>
> __RMSE__
> | Method | Emotion | Gambling | Language | Motor | Relational | Social | WM |
> | :--- | :---: | :---: | :---: | :---: | :---: | :---: | :---: |
> | AM | 0.980±0.002 | 0.974±0.002 | 0.965±0.001 | 0.969±0.003 | 0.976±0.002 | 0.971±0.002 | 0.961±0.001 |
> | VPFM | 0.339±0.003 | 0.313±0.002 | 0.287±0.002 | 0.302±0.003 | 0.317±0.003 | 0.303±0.003 | 0.269±0.003 |
> | [SF]$ ^2$M  | 0.367±0.002 | 0.343±0.003 | 0.313±0.003 | 0.325±0.002 | 0.352±0.003 | 0.330±0.003 | 0.298±0.003 |
> | CFM | 0.314±0.001 | 0.288±0.002 | 0.268±0.001 | 0.280±0.002 | 0.293±0.001 | 0.281±0.002 | 0.249±0.002 |
> | BrainFlow | 0.329±0.002 | 0.303±0.002 | 0.282±0.002 | 0.295±0.002 | 0.307±0.003 | 0.295±0.002 | 0.262±0.002 |
>
> __GCE__
> | Method | Emotion | Gambling | Language | Motor | Relational | Social | WM |
> | :--- | :---: | :---: | :---: | :---: | :---: | :---: | :---: |
> | AM | 0.499±0.003 | 0.473±0.003 | 0.521±0.002 | 0.524±0.005 | 0.442±0.005 | 0.482±0.003 | 0.493±0.007 |
> | VPFM | 0.080±0.009 | 0.060±0.022 | 0.057±0.013 | 0.083±0.016 | 0.071±0.022 | 0.063±0.021 | 0.089±0.012 |
> | [SF]$^2$M | 0.064±0.010 | 0.076±0.017 | 0.077±0.016 | 0.053±0.011 | 0.084±0.026 | 0.084±0.013 | 0.076±0.018 |
> | CFM | 0.095±0.004 | 0.072±0.002 | 0.083±0.002 | 0.083±0.001 | 0.096±0.002 | 0.064±0.002 | 0.075±0.002 |
> | BrainFlow | 0.059±0.003 | 0.071±0.003 | 0.059±0.001 | 0.057±0.002 | 0.070±0.003 | 0.066±0.003 | 0.059±0.003 |
>
>  __LCE__
> | Method | Emotion | Gambling | Language | Motor | Relational | Social | WM |
> | :--- | :---: | :---: | :---: | :---: | :---: | :---: | :---: |
> | AM | 0.503±0.002 | 0.481±0.003 | 0.526±0.002 | 0.526±0.005 | 0.453±0.005 | 0.491±0.003 | 0.502±0.006 |
> | VPFM | 0.089±0.005 | 0.100±0.010 | 0.090±0.006 | 0.089±0.008 | 0.106±0.010 | 0.095±0.009 | 0.095±0.007 |
> | [SF]$^2$M  | 0.093±0.005 | 0.104±0.008 | 0.097±0.007 | 0.085±0.004 | 0.112±0.011 | 0.102±0.006 | 0.101±0.012 |
> | CFM | 0.138±0.002 | 0.125±0.001 | 0.119±0.002 | 0.117±0.001 | 0.128±0.002 | 0.133±0.001 | 0.125±0.002 |
> | BrainFlow | 0.085±0.001 | 0.097±0.002 | 0.084±0.001 | 0.081±0.001 | 0.101±0.002 | 0.090±0.001 | 0.090±0.003 |
> # W5:
> + Thank you for this valuable suggestion. We will __rephrase__ it with “we propose BrainFlow, a reversible generative model designed to parametrize flows between the distribution of SC and the mixed distribution of __FCs in different tasks__”
> # W6-13 ( typos, figures, notations):
> + We will correct these typos and correctly define the notations before being used. We will claim that $q(z)$ is the distribution of the conditioner $z$. We will use $\sigma^2$ in Gaussian kernels
> # Q1:
> + The value of $\epsilon$ we used is fixed to be the largest value in the training set and is not subject-specific. This fixed value enables recovering the prediction of the structural connectivity to the original range.
> # Q2:
> + We first want to correct that the trajectories go __from blue circle to green circle__ instead of from green to blue.
> + Then, As for the transport trajectories, they first go to the center is because both the original velocity $-v_\theta$ and the consensus velocity $v_{con}$ have components that point to center, making the trajectories go to center.
> + Next, the consensus between the point starting from blue circle and that starting from green circle works like *dragging* the center point back to the target position.

---

> > ### Comment · Reviewer_nEq7 · 2025-08-05
> > **Improved quality control required**
> >
> > I thank the authors for addressing the methodological questions in the review. I appreciate the novelty of the proposed methodology. But I am still unconvinced by the performance evaluations with MRI datasets and the relevance to the field; the latter are my areas of expertise.
> >
> > These are my remaining concerns:
> >
> > i) **Poorly written abstract** The abstract, as it stands, is poorly written and does not provide a compelling case for reading the paper. There is a lot of vague jargon like *"ignoring the transform dynamics"* (what transform dynamics?) and *"[ignoring] the SC-FC coupling relationship"* (if previous methods ignored the SC-FC coupling relationship how could _any_ method map from SC-FC?). Moreover, there are unwarranted terms like *"dendritic coupling mechanism"*, which -- as another reviewer has also pointed out -- has nothing to do with diffusion MRI connectivity.
> >
> > Other vague terms in the abstract include *"Since a **spare** of functional connections"* (what is a "spare" of connections?). There is also incomprehensible jargon -- like this phrase -- *"promote the **shared kinetic structures** between multiple FC-to-SC pathways via synchronized coordination"* ("shared kinetic structures" -- this term has no meaning in neuroscience? "synchronized coordination" -- again, redundant phrasing). If the abstract does not convey a clear and concise summary of the key problem statement, the background, and methodological advance, how would a reader understand the relevance of this method for a neuroimaging audience?
> >
> > ii) **Evaluation metrics poorly defined or vague**: The authors use RMSE, LCE and GCE metrics for comparisons. Based on the numbers provided in Table 1, it is clear that other methods like CFM perform better or comparably with BrainFlow for the RMSE. Brainflow seems to perform best with the clustering coefficients, which the authors interpret as: *"BrainFlow more effectively preserves the network’s hierarchical organization at both local and global scales."*
> >
> > However -- to my significant consternation -- I could not find any details provided *at all* anywhere (including the Appendix) about how this clustering was done and how these errors were computed. This a key oversight because better performance on this metric is the major selling point for this method, both with regard to Tables 1 and 3. Not defining this metric anywhere suggests further inadequate quality control.
> >
> > More importantly, there must have been some hyperparameters associated with the clustering? How were these hyperparameters determined? It would be concerning, again, if these hyperparameters were chosen *ad hoc*, and ended up providing an advantage to the BrainFlow method in performance comparisons. Please clarify these details thoroughly.
> >
> > As an aside, performance comparisons in Table 2 are underwhelming -- for 2 out of the 3 datasets, BrainFlow outperforms other methods by ~0.1% when the ground truth classification accuracy is >95%, and upto 99%! Perhaps this table could be moved to the Appendix?
> >
> > iii) **Comparisons with foundational methods**: The authors primarily evaluate other competing flow-based methods and one GAN-based method for mapping SC to FC (or vice versa). But there are a large number of other non-flow based methods in the field. See for example, Zalesky et al (2024; Network Neuroscience). There are a set of standardized evaluation metrics used in such studies (e.g. correlations). While it is not reasonable to expect these comparisons during the limited rebuttal period, a revision must contain such comparisons with baseline methods. As can be seen from Table 3, BrainFlow underperforms poorly relative to the ground truth in the smaller datasets, a limitation that the authors also mention. Simpler methods are particularly useful when data are scarce, and a comparison with simpler, more foundational methods is essential to evaluate the relative utility of the proposed method.

---

> > > ### Author Response · Authors · 2025-08-06
> > >
> > > ### Q1
> > > + We totally agree with the reviewer that the alternating use of concepts from neuroscience, machine learning, and even physics (e.g. kinetic energy/ structure) reduces readability.
> > > + First, we want to clarify that the transform dynamics refers to the progressive transform from one connectivity (SC/ FC) to another type of connectivity (FC/ SC). Regarding "ignoring the SC-FC coupling relationship," we should clarify that previous methods do attempt to establish SC-FC mappings, but they use SC alone as model input and train networks to directly predict FC. In contrast, our method incorporates SC-FC coupling information explicitly by using interpolated states between SC and FC on the manifold space as training inputs, rather than treating them as separate endpoints, thereby capturing the coupling relationship throughout the transformation process rather than only at the endpoints.  As for “dendritic coupling mechanism”, we apologize for the confusing expression, we want to emphasize the “one-to-many” relation between SC and FC with anlogy. To avoid confusion, we will delete these misleading sentences and rephrase it to “we propose BrainFlow, a reversible model designed to parametrize flows between the distribution of SC and the mixed distribution of FCs in different tasks”
> > > + Next, we will rewrite the sentence about the consensus control as “To avoid the cumulative error during flow simulation, we introduce the notion of consensus control to utilize the complementary information s between multiple FC-to-SC pathways, yielding a biologically meaningful underpinning on SC-FC coupling mechanism”
> > >
> > > ### Q2
> > > + We apologize for missing the detailed calculation about the metrics. We will update the explanation about metrics calculation in future version as follows:
> > > + An undirected graph $G = (V, E)$ formally consists of a set of vertices $V$ and a set of edges $E$ between them. An edge $e_{ij}$ connects vertex $v_i$ with vertex $v_j$. For functional connectivity, we only keep the positive edges.
> > >
> > > The local clustering coefficient for undirected graphs can be defined as
> > > $$C_i = \frac{2|\{e_{jk} : v_j, v_k \in N_i, e_{jk} \in E\}|}{k_i(k_i - 1)}$$
> > >
> > > Since any graph is fully specified by its adjacency matrix $A$, the local clustering coefficient for a simple undirected graph can be expressed in terms of $A$ as:
> > > $$C_i = \frac{1}{k_i(k_i - 1)} \sum_{j,k} A_{ij} A_{jk} A_{ki}$$
> > > where:
> > > $k_i = \sum_{j} A_{ij}$
> > >
> > > The global clustering coefficient is defined as:
> > > $$C = \frac{\text{number of closed triplets}}{\text{number of all triplets (open and closed)}}$$
> > >
> > > Since any simple graph is fully specified by its adjacency matrix $A$, the global clustering coefficient for an undirected graph can be expressed in terms of $A$ as:
> > > $$C = \frac{\sum_{i,j,k} A_{ij} A_{jk} A_{ki}}{\frac{1}{2} \sum_{i} k_i (k_i - 1)}$$
> > > where:
> > > $k_i = \sum_{j} A_{ij}$
> > > and
> > > $C = 0 \quad \text{when the denominator is zero}.$
> > > Hence the error rate can be calculated by $\text{error} = \frac{\left| C_{\text{real}} - C_{\text{synth}} \right|}{C_{\text{real}}}.$
> > > There is a tricky case where $C_{\text{real},i}=0$ for _isolated_ node $i$, whose local clustering coefficient is $0$. We mask these nodes as the number of these cases are very small.
> > > + We also emphasize that the calculation of LCE and GCE doesn’t require hyperparameters, making them fair metrics for comparison.
> > > + We will also follow the reviewers’ suggestion to put Table 2 to Appendix.
> > >
> > > ### Q3
> > > + We follow the reviewers' suggestion to add the MLP method in Zalesky et al (2024; Network Neuroscience) to baselines. We also report the eFC-pFC correlation with _BrainFlow_. The correlation metrics of other methods will be displayed on the revised manuscript.
> > > |  | HCP-A | HCP-YA |UKB |
> > > | :--- | :---: | :---: | :---: |
> > > | RMSE | 0.26| 0.24| 0.23 |
> > > | LCE(%) | 8.62 | 12.16 | 12.13 |
> > > | GCE(%) | 5.10| 7.61 | 6.70 |
> > >
> > > __Hypothesis Test__
> > > + | MLP  | HCP-A | HCP-YA |UKB |
> > > | :--- | :---: | :---: | :---: |
> > > | intra | 0.51| 0.50| 0.52 |
> > > | inter | 0.41 | 0.33 | 0.37 |
> > > | p-value | 0.00| 0.00 | 0.00 |
> > >
> > >
> > > + | _BrainFlow_  | HCP-A | HCP-YA |UKB |
> > > | :--- | :---: | :---: | :---: |
> > > | intra | 0.57| 0.56| 0.64 |
> > > | inter | 0.51 | 0.47 | 0.57 |
> > > | p-value | 0.00| 0.00 | 0.00 |
> > >
> > > From the above performance, we admit that we underestimated the potential of simple MLP backbones, which showed consistent performance across different datasets. The inter-intra hypothesis test also suggests that _BrainFlow_ captures individual-specific characteristics of functional connectivity,
> > >
> > > Finally, we wish to concisely highlight that while our evaluation focuses on neuroimaging, BrainFlow is designed as a general and efficient framework for modeling continuous flows on high-dimensional Symmetric Positive-Definite (SPD) manifolds. This makes our method broadly applicable to other scientific domains where data are represented as covariance or connectivity matrices.

---

> > > > ### Comment · Reviewer_nEq7 · 2025-08-07
> > > >
> > > > I appreciate the authors' clarifications. I have only a couple of minor comments:
> > > >
> > > > > For functional connectivity, we only keep the positive edges.
> > > >
> > > > i) This assumption seems a bit problematic because both positive and negative functional connectivity can be mediated by intact structural connectivity.
> > > >
> > > > ii) As defined the LCE and GCE metrics seem not independent, because GCE = sum_i LCE(i). It is not surprising then that a method that performs well on LCE would also perform well on GCE (and perhaps even vice versa). This may be good to clarify upfront in the paper.
> > > >
> > > > > while our evaluation focuses on neuroimaging, BrainFlow is designed as a general and efficient framework
> > > >
> > > > I agree. Which is why I was a bit surprised by not seeing evaluations in other domains; > 80% of the methods in the main paper seem to be general and not restricted to neuroimaging in any way. Perhaps in the revision a few other applications could be showcased.

---

> ### Author Response · Authors · 2025-08-08
>
> Dear Reviewer,
>
> Thank you for your question, we'd like to clarify why we keep the positive edges of FC.
>
> ### Q1.1
> 1. __Definition conflict__
>
> The standard clustering-coefficient formula
> $C_i=\frac{ \sum_{j, k} A_{i j} A_{j k} A_{k i}}{k_i(k_{i-1})}$
> assumes non-negative weights. Negative values can make the numerator negative and the metric no longer interpretable as a “fraction of closed triplets.” For consistency with graph-theory practice [1,2], we set $A_{ij}^- =0$ when computing LCE and GCE.  In addition, clustering coefficient measures the tendency of nodes to form cohesive groups. Positive FC represents cooperative relationships between brain regions, which directly corresponds to the clustering concept. Negative FC represents competitive/anti-correlated relationships, which contradicts the notion of forming cohesive clusters
>
> 2. __No impact on BrainFlow optimisation__
>
> BrainFlow itself receives the signed FC; its transport solver is unchanged. Only the post-hoc metric uses the positive-weight mask, so discarding negatives here cannot bias training or flow estimation.
>
> In addition, we also have tried to use absolute values to calculate the clustering coefficients. However, taking absolute value turns the FC into a fully connected graph, the local topology becomes uniform across subjects. According to the definition of $C = \frac{\text{number of closed triplets}}{\text{number of all triplets (open and closed)}}$, GCE will equal to $0$ constantly.
>
> [1] Fagiolo, G. (2007). Clustering in complex directed networks. Physical Review E—Statistical, Nonlinear, and Soft Matter Physics, 76(2), 026107.
>
> [2] Rubinov, M., & Sporns, O. (2010). Complex network measures of brain connectivity: uses and interpretations. Neuroimage, 52(3), 1059-1069.
>
> ### Q1.2
> + You are correct: by definition $\mathrm{GCE}$ is a __weighted average__ of $\mathrm{LCE}$, where the weight is related to the neighbors of nodes. Hence they are not independent. Our goal was to give both local and global perspectives, not to introduce an independent metric. In the revision, we will state this relationship explicitly in Sec. 4.2 and keep only one of the two values in the main table, with the other moved to the supplement for completeness.
>
> ### Q2
> + Thanks for the reviewers' suggestions. We have included more experiments beside neuroimages. We evalutated _BrainFlow_ on both low-dimension ($2\times 2$) and high-dimensional ($100 \times 100$) SPD data. We tested if the baselines and _BrainFlow_ can keep the SPD property for targets and the Wasserstein distance between generated and real target distribution.
>
> | Mode| SPD ratio (%) (Swiss-roll) | $W_2(\times 10^{-4})$ (Swiss-roll) | SPD ratio (%) (Fig-8) | $W_2(\times 10^{-3})$ (Fig-8) | SPD ratio (%) (High-dim SPD) | $W_2(\times 10^{-2})$ (High-dim SPD) |
> |----------|----------------------------|-------------------------------------|-----------------------|--------------------------------|------------------------------|---------------------------------------|
> | AM       | 100.00   | 9.03    | 100.00                | 1.42                           | 0.00                         | 9.13                                  |
> | VPFM     | 99.96   | 6.28   | 99.99                | 1.93                           | 0.00                         | 7.98                                  |
> | [SF]$^2$M | 99.98  | 6.05  | 99.69                | 1.81                           | 71.98                        | 7.98   |
> | CFM      | 98.84                     | 5.98                                | 99.95                | 1.83                           | 23.78                        | 7.98                                  |
> | RCFM     | 100.00                     | 4.02                                | 100.00                | 1.43                           | 100.00                       | 8.00                                  |
> | *BrainFlow* | 100.00                     | 4.07                                | 100.00                | 1.43                           | 100.00                       | 7.97                                  |
>
> We hope this addresses the reviewer’s concern and will incorporate all the results and discussion in the final version.
>
> We sincerely appreciate your time and efforts of acknowledging our response.

---

> > ### Author Response · Authors · 2025-08-08
> >
> > Dear Reviewer nEq7,
> >
> > We would like to sincerely thank you for the time, effort, and expertise you devoted to reviewing our submission. Your thoughtful feedback, particularly regarding the clarification of the experimental settings, has helped us significantly improve the quality of our work. We are truly grateful for the opportunity to improve our work through this revision process.
> >
> > We hope that our latest revision has adequately addressed your concerns. **If so, we would be most grateful if you would consider updating your score accordingly.**
> >
> > Of course, if you have any remaining concerns or further suggestions, please do not hesitate to let us know, we would be more than happy to address them to the best of our ability.
> >
> > With sincere appreciation,
> >
> > Authors

---

### Decision · Program_Chairs · 2025-09-17

**Decision:**

Accept (poster)

**Comment:**

The authors present a flow matching approach to relate structural and functional connectivity, e.g. in MRI studies, leveraging Riemannian geometry by treating both kinds of connectivity matrices as SPD matrices on the SPD manifold. They also present a theoretical result proving equivalence between flow matching on the SPD manifold and on the Cholesky manifold, which has broader significance.

The paper was reviewed by four expert reviewers.
The reviewers generally agree on the novelty and creativity of the BrainFlow approach, the theoretical contribution, and its potential impact beyond neuroscience. The experimental results for multiple neuroscience datasets were generally well-received by the reviewers. There were multiple concerns regarding the quality of the writing, primarily the abstract and accurate use of terminology, and three reviewers expected a comparison with a broader class of baselines.
Following the rebuttal and discussion, during which the authors clarified details regarding implementation and evaluation, added additional experiments, and promised changes to the writing, the reviewers agreed their concerns were addressed by the detailed explanations and recommended acceptance.

The AC strongly encourages the authors to carefully consider all feedback from the reviewers and implement the promised changes to improve clarity and use of correct terminology, as well as add visualizations requested by the reviewers. The additional baselines and experiments conducted during the rebuttal should also be added to the manuscript to better demonstrate the performance of the approach.